# Task-Aware Data Selection via Proxy-Label Enhanced Distribution Matching for LLM Finetuning

**Hao Cheng**[1*]**, Rui Zhang**[2*]**, Ling Li**[2]**, Na Di**[2]**, Jiaheng Wei**[2,3]**, Zhaowei Zhu**[3]**, Bo Han**[1†]
[1]TMLR Group, Department of Computer Science, Hong Kong Baptist University
[2]The Hong Kong University of Science and Technology (Guangzhou)  [3] D5 Data
{haocheng, bhanml}@comp.hkbu.edu.hk, moeyoo233@gmail.com,
{lli297, ndi595}@connect.hkust-gz.edu.cn,
jiahengwei@hkust-gz.edu.cn, zzw@d5data.ai

## Abstract

Task-specific fine-tuning of foundation models is critically dependent on the quality and relevance of the instruction data. While prevailing data selection methods rely exclusively on instruction instances $X$ to approximate the target distribution, we argue that selection should align with the joint distribution of instructions and task-specific labels $(X, Y)$. However, task-specific labels $Y$ are typically unavailable in practice. To address this, we reformulate the task-specific data selection problem and present a novel pipeline that leverages the reasoning capabilities of large language models (LLMs) to infer proxy labels, thereby facilitating joint distribution alignment. Our approach begins by propagating proxy labels from a small target set to a large, unlabeled source corpus. A two-stage filtering process then removes instances with label noise and refines the subset through distribution alignment. This strategy produces more semantically meaningful and task-aware selections than conventional similarity measures based on $X$ alone. Experimental results show that fine-tuning on a subset of only 10K samples, selected from a pool of 300K, achieves performance competitive or superior to state-of-the-art methods. Code is available at https://github.com/tmlr-group/TADS.

## 1 Introduction

Large language models (LLMs) have demonstrated remarkable capabilities across a wide range of natural language processing tasks, owing to their extensive pretraining on diverse corpora. (Touvron et al., 2023; Achiam et al., 2023; Guo et al., 2025). To adapt these models to specific downstream applications, fine-tuning has become a standard practice. However, the success of fine-tuning critically depends on the quality and relevance of the instruction data, as low-quality or irrelevant data can lead to degraded generalization (Chen et al., 2023; Bukharin & Zhao, 2023). Therefore, selecting high-quality, task-appropriate instruction data is essential for effective model adaptation.

In the field of data selection for LLMs, two primary settings have emerged. The first is task-unspecific data selection (Liu et al., 2023; Chen et al., 2023), which is motivated by the observation that instruction-tuning datasets often contain low-quality or false instructions, which can degrade model performance if used directly in fine-tuning. Moreover, fine-tuning on very large datasets is computationally expensive. Hence, efficient data selection strategies are essential. In contrast, task-specific data selection (Xia et al., 2024; Liu et al., 2024) leverages a small task (*target*) dataset to retrieve the most relevant instruction samples from a *source* pool, aiming at maximizing model performance on particular downstream tasks.

Compared to general task-unspecific data selection, task-specific data selection has drawn growing interest, particularly as practitioners seek to specialize LLMs for specific tasks where limited task

---

*Equal contribution.
†Correspondence to Bo Han (bhanml@comp.hkbu.edu.hk).

(target) examples are available. Most approaches in this area rely exclusively on aligning input features ($X$), typically by measuring embedding similarity or gradient similarity between source and target samples (Xie et al., 2023; Xia et al., 2024; Liu et al., 2024). However, such $X$-only alignment strategies suffer from inherent limitations. For example, consider a target set from the legal domain containing "marital disputes" and "construction contracts." A source sample related to an "cell phone contract" might be selected based on embedding similarity due to semantic overlap with "construction contract." Similarly, if gradient similarity is used—where the model assigns high influence to shared concepts like "contract"—the same irrelevant sample may still be matched. Nevertheless, its actual domain label ("telecom services") does not align with the target label distribution. This discrepancy underscores that input similarity alone is insufficient to ensure task relevance.

To address this, we propose leveraging the joint distribution $P(X, Y)$ rather than marginal distribution $P(X)$ for alignment. Specifically, we harness the reasoning capability of LLMs to infer proxy-labels ($Y$) for the target task, enabling a more semantically meaningful and task-aware selection process. Nevertheless, aligning two random variables is more challenging than aligning only one, and the use of proxy-labels inevitably introduces label noise. In this work, we reformulate the task-specific data selection problem and introduce a novel framework that effectively tackles these issues.

The idea of incorporating auxiliary labels/taxonomy to improve task performance is evident in LLM research. For example, instruction finetuning with the taxonomy of human knowledge has been shown to enhance model generalization (Li et al., 2024). Similarly, in mathematical reasoning, generating intermediate steps (as pseudo-labels) has proven essential for learning complex problem-solving skills (Didolkar et al., 2024). These successes motivate us to leverage proxy-labels to enhance distribution matching for task-specific data selection. Our contributions are threefold:

- We reformulate task-specific data selection as a joint distribution alignment problem and propose a framework that uses LLM-generated proxy-labels to enable alignment.

- We design a two-stage filtering process that first removes out-of-distribution instances prone to open-set label noise, and then refines the subset via distribution alignment.

- Experiments show that fine-tuning on a subset of only 10K samples selected from 300K instruction instances achieves competitive performance compared to state-of-the-art methods.

## 2 RELATED WORKS

**Task-Unspecific Data Selection for Instruction Tuning.** Existing approaches on task-unspecific data selection can be broadly categorized into two types: LLM-free and LLM-based methods. LLM-free approaches (Cao et al., 2023; He et al., 2024) typically leverage handcrafted heuristics or proxy metrics—such as k-NN embedding distances in a feature space, sentence or token-level length or Shapley values to approximate the utility or difficulty of individual samples without invoking large models. In contrast, LLM-based strategies (Chen et al., 2023; Liu et al., 2023; Pang et al., 2024) treat powerful pre-trained language models as automated evaluators to score and filter candidate instruction. Notably, (Lu et al., 2023) propose InsTag, which uses LLM to generate pseudo-labels and refines them for sample selection based on tag statistics. However, InsTag operates in a task-unspecific manner and lacks alignment with a target task distribution. In comparison, our framework is more comprehensive and explicitly incorporates task-aware alignment.

**Task-Specific Data Selection for Instruction Tuning.** Task-Specific Data Selection for Instruction Tuning aims to identify and retrieve the most relevant samples from a large, general-purpose corpus to improve performance on a specific target task, given only a small subset from that target domain. Some methods rely on feature-space similarity: (Yao et al., 2022) select data by ranking candidates based on their similarity to the target set, while (Xie et al., 2023) estimate importance weights in a reduced feature space for resampling. Other methods leverage gradient information: (Xia et al., 2024) score candidates by their maximum gradient similarity to any example in the target set. A more recent approach by (Liu et al., 2024) formulates the problem through the lens of optimal transport, seeking a minimal-cost mapping to align the distributions of the candidate pool and the target set. This method further incorporates diversity-aware regularization to mitigate sample redundancy. However, a common limitation across these methods is that they perform alignment or ranking using only input features, without incorporating the target labels from the target set. This omission may fail to

fully capture task-specific relevance, as the semantic alignment between a candidate's output and the desired target outputs remains unverified.

**Domain Adaptation and Learning with Noisy Labels.** Our approach is further motivated by principles from Domain Adaptation (Wang & Deng, 2018; Zhang et al., 2013) and Learning with Noisy Labels (Natarajan et al., 2013; Song et al., 2022). Specifically, the process of propagating proxy-labels from the target set to the source pool inevitably introduces label noise, a challenge central to learning with noisy labels. Furthermore, even after filtering this noise, a domain shift may persist between the refined source dataset and the target domain, which is a core problem addressed by domain adaptation. Our pipeline is designed to tackle these challenges: we explicitly incorporate mechanisms to identify and filter OOD samples with noisy labels, then align the distributions between the source and target domains to ensure task relevance.

**Our Position.** Our work challenges the prevailing paradigm in task-specific data selection for instruction tuning (e.g., LESS (Xia et al., 2024), TSDS (Liu et al., 2024)), which primarily focuses on aligning marginal input distributions. We argue that true task relevance is defined by the joint distribution of data and labels. By reformulating the selection criteria to incorporate target labels, we unlock a new class of solutions for this problem. The pipeline we introduce is a concrete instantiation of this principle, designed to select data that are not just superficially similar, but semantically congruent with the target task.

## 3 PROBLEM FORMULATION

Define the following concepts:

• **Target Dataset:** Let $\mathcal{T}$ denote a target dataset, where each instance is sampled from a joint distribution $P_{\text{target}}(X, Y_t)$. Here, $X$ represents text instruction instance (typically comprising an instruction, an input, and a response), and $Y_t$ denotes the corresponding domain-specific labels (e.g., for the legal domain, $\mathcal{Y}_t = \{\text{family\_law, contract\_law,... }\}$. [1]

• **Source Dataset:** Let $\mathcal{D} = \bigcup_{k=1}^{K} \mathcal{D}_k$ represent a multi-source dataset comprising $K$ source domains. Each $\mathcal{D}_k$ is sampled from a distribution $P_k(X, Y_k)$. We assume that the support of the target distribution is contained within the union of the supports of the source distributions, that is,

$$\text{supp}(P_{\text{target}}) \subseteq \bigcup_{k=1}^{K} \text{supp}(P_k),$$

which implies that the source domains are relevant to target domains. [2]

**Objective:** The goal is to select a fixed-size subset $\mathcal{S} \subset \mathcal{D}$ where $|\mathcal{S}| = N$ such that such that the empirical distribution $P_S$ approximates $P_{\text{target}}$ as closely as possible. The text instances from $\mathcal{S}$ will then be used to fine-tune a large language model (LLM) with the aim of maximizing performance on the target task.

**Challenges:** The true Labels $Y$ are **unobservable** in both $\mathcal{T}$ and $\mathcal{D}$.

## 4 METHODOLOGY

This section is organized as follows. Section 4.1 describes the use of an LLM for target dataset annotation. Section 4.2 covers label clustering and propagation to the source data. Section 4.3 details the filtering of samples with open-set label noise. Section 4.4 outlines our sampling approach to mitigate domain shift, and Section 4.5 provides an information-theoretic interpretation. An overview of the pipeline is presented in Figure 1.

---

[1] In general, $\mathcal{T}$ may comprise instances from multiple domains (Isobe et al., 2021), in which case $Y_t$ would represent the set of tags across all constituent domains.

[2] This assumption aligns with standard domain adaptation theory (Ben-David et al., 2010) and the common premise of data selection for instruction tuning, where a large source corpus is available. However, our pipeline is also applicable when the source and target domains only partially overlap, such as when source data is limited. See Appendix C for details.

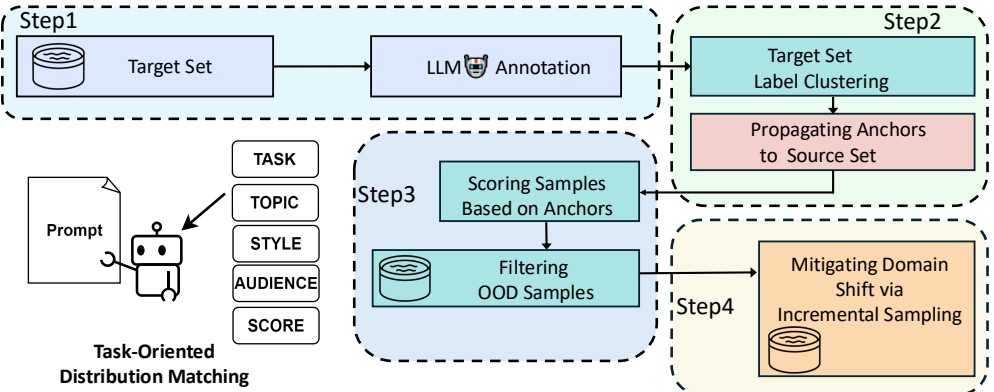

Figure 1: An overview of our 4-step data selection pipeline. We start by using prompt-based LLM annotation to generate domain-specific labels, which are then clustered and propagated to source dataset. Next, an LLM-generated quality score filters out OOD samples with open-set label noise. Finally, incremental sampling mitigates domain shifts between the source and target datasets.

## 4.1 GENERATING PROXY-LABELS BY LLM

We design a structured proxy-label space comprising four key fields: **Task**, **Topic**, **Style**, and **Audience**. These dimensions are designed to capture distinct yet complementary aspects of each instruction instance. Specifically, *Task* refers to the core functional intent, *Topic* reflects the subject, *Style* describes the rhetorical manner, and *Audience* indicates the intended user group. *Task* is modeled as a single-label field since most instructions have a dominant purpose, whereas *Topic*, *Style*, and *Audience* are multi-label fields to account for multifaceted instruction characteristics. We then prompt a pre-trained LLM with a structured summarization template (see Appendix B) to annotate each instruction with tags in four semantic fields—*Task*, *Topic*, *Style*, and *Audience*.

This semantic decomposition results in modular and interpretable representations that enable structured downstream processing, including clustering, alignment, and semantic similarity estimation. The use of a single-label format for *Task* provides a well-defined functional anchor for each instance, while multi-label modeling for *Topic*, *Style*, and *Audience* captures the multifaceted nature of instructional content. Free-form, descriptive phrase tags allow the representation space to remain open-ended and continuous, avoiding constraints imposed by fixed label taxonomies and facilitating compatibility with embedding-based semantic matching.

It is worth noting that our defined four **general** label domains (*Task*, *Topic*, *Style*, *Audience*) are used to demonstrate our framework's flexibility. In practice, **general** label domains are applied when target knowledge is scarce, for instance, when only a set of target samples is available without clear target details. Conversely, when prior target knowledge exists, more **specific** label domains can be adopted. For example, using a domain-specific label such as "mathematical problem-solving skills" for math reasoning tasks has been shown to improve exemplar quality (Didolkar et al., 2024), indicating that more precise domain definitions generally lead to better data selection. In this paper, we use general label domains in experiments to emphasize our pipeline's efficiency.

## 4.2 PROXY-LABEL CLUSTERING AND PROPAGATION

To bridge the semantic gap between source domains and our LLM-generated labels, we employ a robust methodology centered on clustering and semantic propagation. First, we encode each label into a dense vector space using a sentence-level embedding model, processing each field (e.g., *Topic*, *Style*) independently to preserve their unique semantic characteristics.

**Proxy Label clustering** We apply $k$-means clustering (with $k = 100$) to these embeddings, which effectively groups semantically similar tags into distinct clusters (Figure 3 demonstrates the robustness to $k$). This process is fundamental because it accomplishes two key objectives: it abstracts away the significant lexical variation inherent in LLM-originated phrases and condenses their broad diversity

into a compact set of semantic centroids. These centroids function as semantic anchors, providing a stable and noise-reduced representation of the entire target domain's conceptual landscape.

**Propagation** We project samples from the source domains into the same embedding space and compute their cosine similarity against all semantic anchors. This allows for a soft, interpretable matching process. We then assign the most relevant anchors—selecting the top-1 for Task and the top-3 for Topic, Style, and Audience, to each source sample. This selective assignment is not a hard label transfer but a test-guided semantic projection, enabling a nuanced and measurable integration of the target domain's characteristics into the source.

## 4.3 Filtering OOD Samples with Label Noise

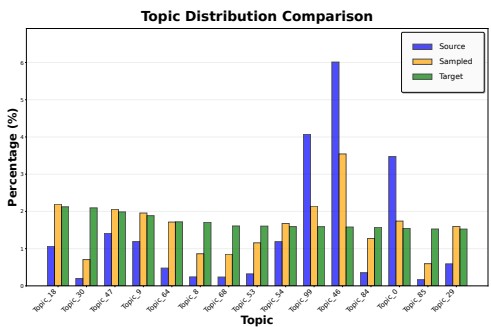 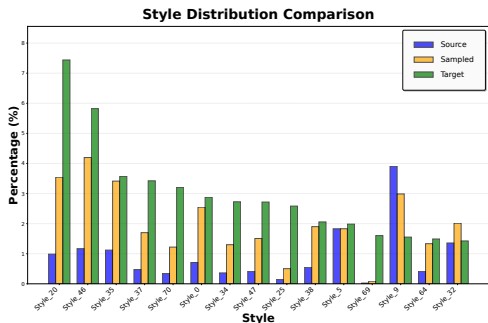

Figure 2: These figures visualize the results of our incremental sampling method. The left and right panels show the matching results for the *Topic* and *Style*, respectively. The anchors are ranked from highest to lowest count of target set (left to right) for clarity, and the top 15 anchors are shown to enhance readability.

The propagation of target domain anchors to source domains inevitably introduces open-set label noise where irrelevant instructions are assigned to an target label. To ensure data quality, filtering these mismatched samples becomes essential. In learning with noisy labels, a common detection strategy leverages the memorization effect of deep neural networks (DNNs), wherein DNNs learn clean patterns before memorizing noisy ones (Arpit et al., 2017). By training a DNN for a few epochs and ranking samples by loss values, high-loss instances are typically identified as noisy—an approach effective for both closed-set and open-set noise (Jiang et al., 2017; Han et al., 2018; Li et al., 2020; Sachdeva et al., 2021).

However, this method presents limitations in our multi-field setting: training separate DNNs is computationally expensive and lacks generalization capability. Each new task or label space change would necessitate retraining, creating scalability bottlenecks. To overcome these constraints, we leverage the robust reasoning capabilities of large language models (LLMs). Specifically, we employ LLMs to score each sample based on its semantic alignment with assigned anchors, evaluating the quality and relevance of the assigned anchors (see Appendix B for prompt details). Samples falling below a predefined threshold (e.g., score < 6) are identified as out-of-distribution (OOD)—indicating open-set label noise—and subsequently removed from the source corpus (Appendix C has further explanations).

## 4.4 Incremental Sampling for Mitigating Domain shifts

In practice, after removing OOD samples, a simple approach is to randomly sample a fixed number of instances (e.g., 10,000) from the source domain for fine-tuning. However, even after OOD filtering, a distribution shift may persist between the remaining source data and the target set. This section presents a method to mitigate such domain shifts more effectively.

Leading works in domain adaptation (Zhang et al., 2013; Zhao et al., 2019) theoretically and empirically demonstrate that target shift (or prior probability shift) is a common and critical type of distribution discrepancy in real-world scenarios and suggest that aligning for target shift is often more advantageous for robustness and effectiveness than aligning for covariate shift. Consequently,

our pipeline prioritizes mitigating target shift between the source and target domains. The implicit assumption of target shift is that the conditional distribution $P(X|Y)$ remains stable across the target and cleaned source domains, while the label distribution $P(Y)$ varies. Note that the label space here corresponds to the anchors (clusters) defined in Section 4.2 , not the labels/tags generated by the LLM in Section 4.1.

**Sampling Objective:** Let $P^*(Y)$ be the empirical anchor distribution of the target dataset and $\hat{P}(Y)$ be the empirical anchor distribution of the sampled source dataset. To mitigate label (anchor) shift, our objective is to minimize $||P^*(Y) - \hat{P}(Y)||_1$.

Given a sampling budget $N$, we propose an incremental sampling method outlined in Algorithm 1. At each step, we compute the label gap and select a training instance associated with the label with the largest positive gap (Lines 4-5, Algorithm 1). Because each instance contains a fixed number of labels per field (e.g., one for *Task*, three for *Topic*), all candidates contribute equally to the label counts. The process iterates until the budget is exhausted or no viable candidates remain, producing a subset whose empirical distribution $\hat{P}(Y)$ closely approximates the target distribution $P^*(Y)$.

Algorithm 1 is applied independently to each field (e.g., *Task* or *Topic*). For each selected sample, we update the counts for all associated labels. This introduces a natural balancing effect across the label space, accelerating convergence toward $P^*(Y)$, especially in dense semantic fields where tags frequently co-occur. Figure 2 presents the matching results, demonstrating that incremental sampling produces a source subset whose anchor distribution aligns with the target distribution. Although perfect matching is rarely achievable due to limited sample sizes for certain classes in the source dataset and the sampling budget, our incremental sampling method effectively reduces the distribution gap between the source and target domains. A key implication of this method is its flexibility. Since alignment is performed per field, we can choose to align by *Task*, *Topic*, or other fields based on the specific requirements of the downstream application.

---

**Algorithm 1 Distribution Matching via Incremental Sampling**

---

**Require:** source pool $\mathcal{D}$, target set $\mathcal{T}$, sampling budget $N$
 1: Compute target label distribution $P^*(Y)$ from $\mathcal{T}$
 2: Initialize empirical distribution $\hat{P}(Y) \leftarrow \mathbf{0}$ and selection set $\mathcal{S} \leftarrow \emptyset$
 3: **while** $|\mathcal{S}| < N$ **do**
 4:     Compute label-wise gap: $g(y) = P^*(y) - \hat{P}(y)$
 5:     Identify label $y^* = \arg\max_y g(y)$ such that unused candidates with tag $y$ exist
 6:     **if** no valid $y^*$ found **then**
 7:         **break**
 8:     **end if**
 9:     Select candidate $x^*$ with tag $y^*$
10:     $\mathcal{S} \leftarrow \mathcal{S} \cup \{x^*\}$; update $\hat{P}(Y)$ using labels from $x^*$
11: **end while**
12: **return** Sample set $\mathcal{S}$

---

## 4.5 INFORMATION-THEORETIC EXPLANATION

We explain the benefits of incorporating label information in the dataset selection process from an information-theoretic perspective. Let $T$ be a random variable representing the **target domain**, which is charactirized by $P_{\text{target}}(X, Y_t)$. The outcome of the selection algorithm is a **subset** $\mathcal{S} \subset \mathcal{D}$. The instances and their associated labels of $\mathcal{S}$ are described by the random variables $X_{\mathcal{S}}$ and $Y_{\mathcal{S}}$.

The mutual information between the task and the selected data can be decomposed as:

$$I(T; (X_{\mathcal{S}}, Y_{\mathcal{S}})) = I(T; X_{\mathcal{S}}) + I(T; Y_{\mathcal{S}} \mid X_{\mathcal{S}})$$

This decomposition reveals two distinct sources of information:

- $I(T; X_{\mathcal{S}})$ quantifies the information about the task $T$ contained in the *input features* $X_{\mathcal{S}}$ of the selected subset.
- $I(T; Y_{\mathcal{S}} \mid X_{\mathcal{S}})$ captures the *additional information* about $T$ provided by the *labels* $Y_{\mathcal{S}}$ of the subset, given the inputs $X_{\mathcal{S}}$.

Conventional task-specific data selection methods in LLM (Xia et al., 2024; Liu et al., 2024), which rely exclusively on the similarity between input features (i.e., $X_\mathcal{S}$), primarily aim to maximize the first term, $I(T; X_\mathcal{S})$. However, they inherently ignore the second term, $I(T; Y_\mathcal{S} \mid X_\mathcal{S})$. This constitutes a limitation, since instances with semantically similar $X$ may be associated with labels $Y$ that are irrelevant or even contradictory to the target task $T$. In contrast, our proposed method matches the joint distribution between the selected data and the target domain, which implicitly maximize the complete objective $I(T; (X_\mathcal{S}, Y_\mathcal{S}))$.

## 5 EXPERIMENTS

### 5.1 EXPERIMENTS SETUP

We utilize three pretrained LLMs across different stages of our pipeline: **Qwen2.5-7B-Instruct** (Yang et al., 2024) is used for annotation and scoring, **BGE-M3** (Chen et al., 2024) for embedding and clustering, and **LLaMA-3.1-8B** (Grattafiori et al., 2024) for supervised instruction tuning on the selected subsets using **LoRA** (Hu et al., 2022).

Table 1: Instruction tuning data pool and evaluation benchmarks

**Instruction Tuning Data Pool**

| Dataset | Flan V2 | Open-Assistant 1 | WizardLM | Dolly | Stanford Alpaca | Total |
|---|---|---|---|---|---|---|
| **Training Size** | 100K | 33K | 100K | 15K | 52K | **300K** |

**Evaluation Benchmarks**

| Benchmark | MMLU | TruthfulQA | GSM8K | BBH | TyDiQA | Total |
|---|---|---|---|---|---|---|
| **Test Size** | 14,042 | 790 | 1,319 | 6,511 | 5,077 | **27,739** |
| **Capability** | Factuality | Truthfulness | Reasoning | Reasoning | Multilinguality | |

**Data Pool** Our data pool comprises two components: a source corpus collected from five widely-used instruction tuning datasets—**Flan V2** (Longpre et al., 2023), **Open-Assistant 1** (Köpf et al., 2023), **WizardLM** (Xu et al., 2023), **Dolly** (Databricks, 2023), and **Stanford Alpaca** (Taori et al., 2023)—and a set of evaluation benchmarks. Evaluation is conducted on five standard alignment benchmarks, each targeting a distinct model capability: **MMLU** (Hendrycks et al., 2020), **TruthfulQA** (Lin et al., 2021), **GSM8K** (Cobbe et al., 2021) and **BBH** (Suzgun et al., 2022), and **TyDiQA** (Clark et al., 2020). Detailed training set statistics and the sizes of evaluation sets are provided in Table 1. These benchmarks also serve as our alignment targets: for each dataset, we sample 20% of the test set to form a target set; all target sets are then combined to guide our subset selection. The remaining 80% of each test set is reserved for final evaluation.

**Baselines:** We compare our method against a set of representative baselines commonly used in instruction tuning and data selection:

- **Vanilla Base Model**: The base model evaluated in a zero-shot setting, without any supervised fine-tuning. This serves as a reference point to measure the impact of instruction tuning.

- **Full (300K)**: The entire 300K training set is used for instruction tuning.

- **Completion Length**: Following (Zhao et al., 2024), we rank all samples by the total number of tokens in the prompt-response pair. The top 10K longest samples are selected under the assumption that longer responses carry higher information density.

- **k-NN**: Each sample is scored by its average distance to the $k$ nearest neighbors in a sentence embedding space (Reimers & Gurevych, 2019). Higher distances indicate semantic uniqueness, and the 10K most isolated samples are selected.

- **BM25**: A retrieval metric which considers term frequency and inverse document frequency.

- **RDS+** (Ivison et al., 2025): A representation-based retrieval method which uses weighted mean pooling of pretrained LLM hidden states.

Table 2: Performance comparison across alignment benchmarks. All the methods select 10K samples from the source except the method of Full (300K) and Vanilla Base Model. For our align based method, we use *min score* $\geq 7$ to filter samples in this table, as described in Section 4.3. For each benchmark, top-performing score is shown in bold, while second-best score is underlined. We also record *mean* and *std* for selection methods.

| Methods | MMLU | TruthfulQA | GSM | BBH | TyDiQA |
|---|---|---|---|---|---|
| Vanilla Base Model | 64.3 | 32.8 | 51.0 | 54.8 | 22.7 |
| Full (300k) | 63.6 | 44.1 | 49.0 | 57.9 | **66.4** |
| Completion Length | $63.3_{0.04}$ | $7.7_{1.1}$ | $54.3_{1.43}$ | $61.0_{0.65}$ | $62.3_{0.45}$ |
| k-NN-10 | $61.9_{0.37}$ | $41.1_{0.70}$ | $52.0_{0.41}$ | $\underline{61.3_{0.49}}$ | $61.3_{0.05}$ |
| Random | $63.7_{0.24}$ | $29.1_{3.41}$ | $54.0_{1.22}$ | $60.2_{0.33}$ | $60.5_{0.33}$ |
| BM25 | $63.2_{0.03}$ | $25.2_{2.77}$ | $51.3_{0.20}$ | $58.8_{0.28}$ | $62.0_{0.29}$ |
| RDS+ | $63.8_{0.12}$ | $3.6_{0.04}$ | $53.8_{0.20}$ | $59.4_{0.21}$ | $60.3_{0.08}$ |
| LESS | $63.3_{0.04}$ | $35.1_{1.41}$ | $56.1_{0.61}$ | $61.1_{0.54}$ | $\underline{64.0_{0.54}}$ |
| TSDS | $63.6_{0.08}$ | $44.1_{0.69}$ | $50.0_{1.22}$ | $\mathbf{62.1_{0.22}}$ | $63.5_{0.54}$ |
| Align_topic | $\mathbf{64.5_{0.12}}$ | $46.9_{0.29}$ | $57.0_{0.22}$ | $60.2_{0.29}$ | $58.6_{0.61}$ |
| Align_style | $64.2_{0.08}$ | $\mathbf{47.2_{0.12}}$ | $\mathbf{57.8_{0.61}}$ | $58.9_{0.29}$ | $59.3_{0.45}$ |
| Align_task | $\underline{64.4_{0.16}}$ | $46.5_{0.53}$ | $\underline{57.5_{0.41}}$ | $58.9_{0.57}$ | $59.3_{0.90}$ |
| Align_audience | $64.4_{0.21}$ | $\underline{47.1_{1.22}}$ | $55.8_{1.13}$ | $58.5_{0.73}$ | $60.2_{1.05}$ |

Table 3: Performance comparison for our align-based method with respect to *min score*.

| Methods | MMLU | TruthfulQA | GSM | BBH | TyDiQA |
|---|---|---|---|---|---|
| *Align-based selection, min score $\geq 7$* | | | | | |
| Align_topic | 64.5 | 46.9 | 57.0 | 60.2 | 58.6 |
| Align_style | 64.2 | **47.2** | **57.8** | 58.9 | 59.3 |
| Align_task | 64.4 | 46.5 | $\underline{57.5}$ | 58.9 | 59.3 |
| Align_audience | 64.4 | $\underline{47.1}$ | 55.8 | 58.5 | **60.2** |
| *Align-based selection, min score $\geq 6$* | | | | | |
| Align_topic | 65.0 | 24.7 | 55.5 | $\underline{61.9}$ | 57.0 |
| Align_style | **65.1** | 32.1 | 55.0 | **62.5** | 58.9 |
| Align_task | 65.0 | 38.0 | 54.0 | 59.8 | 58.1 |
| Align_audience | 64.9 | 42.6 | 55.0 | 59.7 | 59.1 |
| *Align-based selection, min score $\geq 5$* | | | | | |
| Align_topic | $\underline{65.1}$ | 36.2 | 52.5 | 61.4 | 57.6 |
| Align_style | 64.9 | 31.2 | 55.0 | 60.4 | $\underline{59.4}$ |
| Align_task | 64.9 | 35.6 | 54.5 | 60.0 | 57.7 |
| Align_audience | 65.1 | 39.2 | 55.0 | 60.6 | 58.0 |

- **Random**: A simple baseline that randomly selects 10K examples from the data pool.

- **LESS** (Xia et al., 2024): A gradient-based data selection framework that estimates the influence of training samples using gradient similarity to few-shot target examples.

- **TSDS** (Liu et al., 2024): A task-specific data selection framework that aligns the source distribution with a small representative set from the target task. TSDS formulates data selection as an optimal transport problem with a diversity-promoting regularizer based on kernel density estimation. We use the target set to estimate the task distribution and select 10K training samples accordingly, ensuring parity with our proposed method.

Table 4: Performance comparison across alignment benchmarks with 1K/5K selection size.

| Methods | MMLU | TruthfulQA | GSM | BBH | TyDiQA |
|---|---|---|---|---|---|
| Random (1K) | 64.0 | 3.8 | 49.0 | 54.2 | 56.4 |
| Align_style (1K) | 64.2 | 6.8 | 50.5 | 57.7 | 54.5 |
| Random (5K) | 63.7 | 3.5 | 52.0 | 59.4 | 58.7 |
| Align_style (5K) | 63.9 | 32.4 | 52.0 | 61.0 | 59.7 |

## 5.2 OPENLLM LEADERBOARD EVALUATION RESULTS

Following the OpenLLM Leaderboard evaluation protocol, we adopt *exact match* (EM) as the scoring metric for MMLU, TruthfulQA, GSM8K, and BBH. For TyDiQA, we use the 1-shot F1 score as reported by the leaderboard.

**Results Analyses:** As shown in Table 2 and Table 3, our method achieves competitive or superior performance compared to SOTA methods (LESS, TSDS) on task-specific data selections for the MMLU, TruthfulQA, GSM8K and BBH benchmarks. However, our approach underperforms on TyDiQA. We hypothesize that this is because TyDiQA is a multilingual dataset, and the label fields we designed may not adequately capture its inherent characteristics. Consequently, samples related to TyDiQA in the source pool were likely filtered out by our selection algorithm.

From Table 3, the *min score* threshold also has impacts on performance. Analogous to the problem of learning with noisy labels (Jiang et al., 2017; Han et al., 2018), a higher threshold risks retaining too few clean (relevant) samples, while a lower threshold risks involving too many irrelevant samples.

It is important to note that even though we use four label domains for performing experiments, our intention is not to suggest empirically testing all label domains for each new task. The primary purpose of presenting four domains is to illustrate the inherent flexibility of our pipeline. The results in Table 2 demonstrate that selecting any of these domains yields strong performance on MMLU, TruthfulQA, and GSM benchmarks compared to baseline methods. For tasks with more specific information, a custom domain can also be designed, as discussed in Section 4.1. We further evaluate our pipeline with limited data. As shown in Table 4, our method achieves good performance even with smaller selection sizes (1K and 5K), demonstrating its effectiveness.

## 5.3 ABLATION STUDIES

Table 5 presents the results of our ablation study on the contributions of OOD sample filtering and incremental sampling. We compare three configurations:

**Incremental Sampling only:** This baseline applies incremental sampling to the entire 300K dataset after label propagation, without any OOD filtering; **OOD Filtering only:** This variant filters the dataset by retaining only samples with an LLM-assigned *min score* $\geq 6$ , followed by random sampling; **Complete Method:** This is our full pipeline, which first applies OOD filtering (*min score* $\geq 6$) and then performs incremental sampling on the filtered subset. For each benchmark, results are reported under the best-performing label field. The result clearly indicates that combining both techniques is most effective and leads to superior performance.

Table 5: Evaluating individual contributions of OOD sample filtering and incremental sampling.

| Methods | | MMLU | TruthfulQA | GSM | BBH | TyDiQA |
|---|---|---|---|---|---|---|
| OOD Filtering | Incremental Sampling | | | | | |
| ✗ | ✓ | 64.3 | 3.5 | 55.0 | 49.1 | 60.5 |
| ✓ | ✗ | 64.8 | 42.1 | 55.0 | 61.9 | 56.4 |
| ✓ | ✓ | 65.1 | 42.6 | 55.5 | 62.5 | 59.1 |

To further verify the necessity of joint alignment on $P(X, Y)$ over input-only alignment on $P(X)$, and to demonstrate the advantage of our design, we conduct the following experiments:

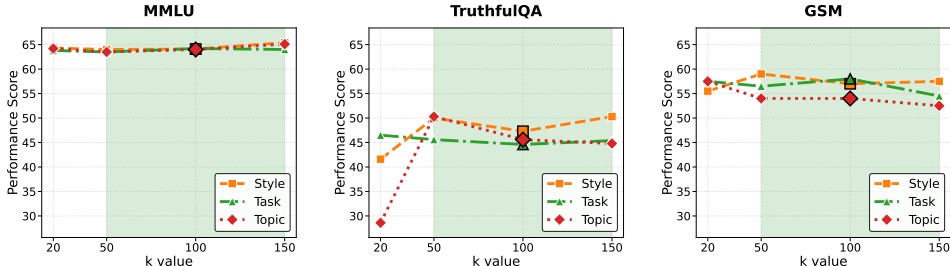

Figure 3: Performance of our pipeline with respect to the number of clusters $k$ in K-means clustering.

**Adding semantic information to an existing selection pipeline.** After LLM annotation, we insert the proxy label information into each instruction and apply the standard RDS+ selection pipeline; **Applying our pipeline directly to inputs** $X$. We remove the LLM-annotation step entirely. Target instructions are clustered into anchors, source samples are propagated according to anchor–source embedding similarity, and we then perform selection and incremental sampling; **Replacing LLM-based OOD filtering with embedding-similarity filtering.** Instead of using LLM-assigned scores, we compute cosine similarity between source and anchor embeddings, filter out low-similarity samples, and then apply our selection and incremental sampling procedure.

As shown in Table 6: (1) The direct integration of semantic information (RDS+ with semantic info) fails to yield consistent improvements over RDS+, indicating that explicit distribution alignment is necessary to utilize semantic-field information effectively; (2) Our pipeline outperforms its direct application to $X$, confirming that the performance gains stem from joint alignment on $P(X, Y)$ rather than from clustering and sampling alone; (3) Compared to similarity-based OOD filtering, our method demonstrates that LLM-based judgments provide more reliable filtering.

Table 6: Comparison of alternative designs for our joint alignment.

| Methods | MMLU | TruthfulQA | GSM | BBH | TyDiQA |
|---|---|---|---|---|---|
| RDS+ | 63.6 | 3.5 | 54.0 | 59.1 | 60.2 |
| RDS+ (with semantic information) | 63.7 | 5.4 | 53.5 | 60.7 | 59.7 |
| Applying our pipeline directly to $X$ | 63.4 | 40.8 | 54.0 | 59.9 | 58.2 |
| Our pipeline with similarity-based OOD filtering | 63.0 | 28.3 | 51.0 | 57.4 | 61.6 |
| Our full pipeline (align topic, min score:7) | 64.6 | 46.4 | 57.0 | 60.0 | 59.3 |

We also evaluate the impact of the K-means cluster count $k$ on our pipeline's performance in Figure 3. The results show consistent performance for $k$ between 50 and 150 across datasets and label domains. However, a smaller $k$ can lead to higher variance, as seen on TruthfulQA. For robustness and reproducibility, we fix $k = 100$ in all experiments.

The Appendix includes more experiments and analyses including the robustness of our pipeline to variations in label quality, deeper investigation into the connection between our pipeline and the fields of Domain Adaptation (DA) and Learning with Noisy Labels (LNL), etc.

## 6 CONCLUSION

In this paper, we revisit task-aware data selection by formulating it as a joint alignment of both input features $(X)$ and task-specific labels $(Y)$. This paradigm shift moves beyond input-only alignment to naturally address real-world challenges like label noise and domain shift. By integrating techniques from noise-robust learning and domain adaptation, we propose a pipeline that effectively mitigates these issues. The primary contribution is not merely the specific pipeline but the reconceptualization of the problem itself. Framing it as joint distribution alignment unlocks a broader solution space for future research. Our work serves as an initial instantiation of this perspective.

## ACKNOWLEDGEMENTS

HC and BH were supported by NSFC General Program No. 62376235, Guangdong Basic and Applied Basic Research Foundation Nos. 2024A1515012399, and HKBU CSD Departmental Incentive Scheme.

## ETHICS STATEMENT.

Our work focus on task-specific data selection for fine-tuning large language models (LLMs). All datasets and models used are publicly available; therefore, no intellectual property or personal privacy concerns are involved.

## REPRODUCIBILITY STATEMENT.

We have released our code at `https://github.com/tmlr-group/TADS`.

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

## APPENDIX ARRANGEMENT

The Appendix is arranged as follows:

- **Section A:** Statement of the Use of Large Language Models (LLMs) in writing.
- **Section B:** Prompt templates employed in our proposed pipeline.
- **Section C:** Deeper analysis of the connections between our pipeline and the fields of Domain Adaptation (DA) and Learning with Noisy Labels (LNL).
- **Section D:** Additional experimental results and visualizations.
- **Section E:** Detailed experimental settings.

## A  STATEMENT OF THE USE OF LARGE LANGUAGE MODELS

In this paper, DeepSeek was used solely to check grammar and polish writing style. No scientific material or core research content was generated by any Large Language Model (LLM).

## B  LLM PROMPT TEMPLATE

The following is the prompt template we used in the task-specific label generation in Section 4.1.

---

**Prompt Template for LLM Annotation**

**<System Prompt>**: You are a helpful assistant.
**<User Prompt>**: [Background Information]
Instruction: [Instruction]
Input: [Input]
Response: [Response]
Please strictly output in JSON format and generate rich, high-quality summaries suitable for LLM. Summarize the entire conversation **as a single summary**, not per message or per role.

Requirements:
1. Task: Describe the single most appropriate task type from the background as a short sentence or detailed multi-phrase descriptor (e.g., "Answering open-domain factual questions based on user input").
2. Style: List the top 3 most relevant styles or tones as detailed phrases or brief sentences (e.g., "Uses a formal and professional tone suitable for academic writing").
3. Topic: List 3 key topics covered in the background.
4. Audience: List the top 3 intended audiences.

Only return a single JSON object as shown below:

```
{
    "Task": "<short sentence or descriptive phrase>",
    "Style": [
        "<descriptive phrase>",
        "<descriptive phrase>",
        "<descriptive phrase>"
    ],
    "Topic": [
        "<short sentence>",
        "<short sentence>",
        "<short sentence>"
    ],
    "Audience": [
        "<descriptive phrase>",
        "<descriptive phrase>",
        "<descriptive phrase>"
    ],
}
```

---

The following is the prompt template we used in Section 4.3 to score the source sample based on its assigned anchors.

---

**Prompt Template for Scoring Source Samples**

**<System Prompt>**: You are a helpful assistant.
**<User Prompt>**: You are an expert evaluator. Please evaluate the following sample based on these criteria:
– Completeness (1-10): How complete is the response?
– Information Richness (1-10): How much useful information does it contain?
– Rarity (1-10): How unique or rare is this type of content?
– Complexity (1-10): How complex is the task/content?
Sample to Evaluate: [sample_format]
Associated Tags:
– Task: [Assigned Task anchor]
– Style tags: [Assigned Style anchor]
– Topic tags: [Assigned Topic anchor]
– Audience tags: [Assigned Audience anchor]
Use these tags as reference points for your scoring decisions. Consider ALL the tags in each category when evaluating. Please respond with ONLY a JSON object in this exact format:

```
{
    "Completeness": "<1-10>",
    "Information Richness": "<1-10>",
    "Rarity": "<1-10>",
    "Complexity": "<1-10>",
    "Overall Score": "<1-10>"
}
```

---

Note that each anchor may contain many domain-specific tags, as the anchors are generated by clustering. Therefore, the anchor description in the prompt template above is summarized by LLM, which extracts the 20 most representative keywords or key phrases for each anchor.

## C  DEEPER ANALYSES OF OUR PIPELINE

In this section, we provide answers and explanations for the following two questions:

1. Why our approach in Section 4.3 can be regarded as a proxy for filtering label noise?

2. Can our pipeline be applied to scenarios where the target and source domains only partially overlap?

For the first question, our approach in Section 4.3 can be viewed as a proxy for filtering label noise because it mirrors a established technique in learning with noisy labels. In traditional classification, models tend to be less confident about mislabeled samples compared to clean samples, resulting in a low maximum softmax probability (the "confidence score"). This score is a common signal for identifying potential label noise (Hendrycks & Gimpel, 2016). Instead of training a model to obtain a confidence score, we leverage the reasoning capability of an LLM. The LLM assesses a sample's relevance to its most similar anchors and assigns a score, which functions as a semantics-aware confidence score. Thus, our method provides a direct analog to confidence-based filtering without the need for model training.

For the second question, we use Figure 4 to demonstrate our pipeline's applicability to scenarios where the target and source domains only partially overlap.

- In Figure 4 (a), the support of the target domain is a subset of the source domain. Our OOD filtering step first produces a filtered source dataset with a support that approximately matches the target. Subsequent incremental sampling then aligns the distribution of the selected dataset with the target distribution.

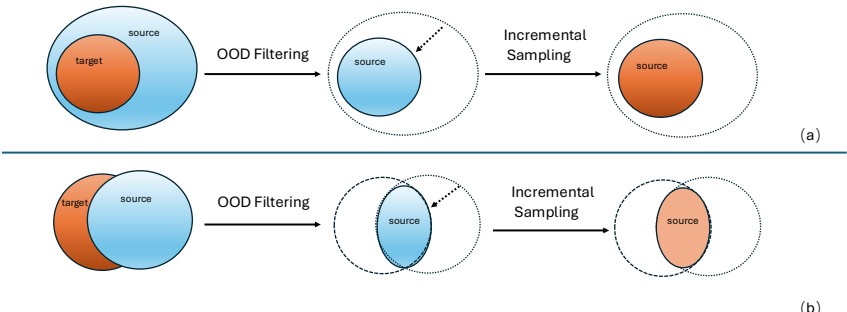

Figure 4: Illustration of how our pipeline aligns the target distribution by sampling from the source. The upper figure (a) shows the case where the support of the target distribution is a subset of the source support. The lower figure (a) shows the case of partial overlap between the target and source distributions.

Table 7: Performance comparison of different data selection approaches for fine-tuning the Mistral-7B-v0.3 model. Each approach selects 10K samples. For each benchmark, top-performing score is shown in bold, while second-best score is underlined.

| Methods | MMLU | TruthfulQA | GSM | BBH | TyDiQA |
|---|---|---|---|---|---|
| Vanilla Base Model | 59.7 | 30.0 | 38.0 | 49.1 | 54.4 |
| LESS | 59.6 | 32.0 | 38.5 | 51.9 | **59.2** |
| TSDS | 59.4 | **37.0** | 41.5 | **57.4** | 56.4 |
| Ours (Align-topic) | 60.3 | 35.4 | 44.0 | 52.8 | 53.9 |
| Ours (Align-task) | **60.4** | 36.6 | **46.0** | 53.0 | 54.2 |
| Ours (Align-style) | 60.0 | 34.3 | 40.5 | 51.0 | 55.3 |
| Ours (Align-audience) | 60.3 | 35.1 | 43.5 | 51.5 | 55.1 |

- In Figure 4 (b), after OOD filtering, the support of the filtered source dataset becomes a subset of the target domain. Incremental sampling then matches the distribution of the selected data to the distribution of the partial target (i.e., the shared portion), confirming that our pipeline successfully selects task-relevant samples.

Notably, the scenario in (b) corresponds to the open-set Domain Adaptation (DA) setting. Similar to traditional open-set DA methods (Panareda Busto & Gall, 2017; Liu et al., 2019), which focus on aligning shared classes while rejecting unknown ones, our pipeline inherently handles this case by filtering and aligning only the overlapping regions.

# D  ADDITIONAL EXPERIMENTAL RESULTS AND EXAMPLES.

## D.1  FINETUNING ON OTHER FOUNDATION MODELS

In the main paper, we fine-tune the Llama-3.1-8B model on our selected data. In this section, we instead fine-tune Mistral-7B-v0.3 to demonstrate the generalizability of our selection pipeline. The results in Table 7 demonstrate that our method achieves competitive performance on the Mistral model. Specifically, Align-task achieves top-1 or top-2 performance across all benchmarks except TyDiQA.

## D.2  EXAMINING THE LLM ANNOTATION QUALITY

We conduct a thorough evaluation of the quality of proxy labels generated by the LLM, assessing both precision (whether a label correctly describes its corresponding instruction) and consistency (whether the same label is applied to semantically similar instructions). Our examining methodology follows the framework established by Lu et al. (2023).

- **For precision evaluation:** We randomly sampled 500 instruction-label pairs. GPT-4 was asked to assess if the label adequately describes the instruction. From these, the first 50 pairs were selected for human evaluation by three independent annotators.

- **For consistency evaluation:** We randomly sampled 500 labels, ensuring each label was associated with at least two distinct instructions. GPT-4 was asked to determine if the instructions for a given label were semantically consistent. Similarly, the first 50 sets of instructions were assessed by three human annotators.

We calculated the agreement between raters using Cohen's Kappa (for pairwise agreement between human and GPT-4) and Fleiss' Kappa (for inter-annotator agreement among humans). The results are presented in Table 8.

Table 8: Annotation Quality Comparison

| Metric | GPT-4 Annotation | Human Annotation | Agreement | |
|---|---|---|---|---|
| | | | Human-Human | Human-GPT |
| Tag Precision | 0.958 | 0.94 | 0.4823 | 0.7899 |
| Tag Consistency | 0.856 | 0.9 | 0.73 | 0.736 |

As shown in Table 8, both GPT-4 and human evaluations indicate high precision and consistency for the proxy labels. The Human-GPT agreement scores (exceeding 0.7) indicate solid alignment between human and LLM judgments.

### D.3    LLM score consistency

To assess the reliability of LLM-based OOD filtering, we evaluate score consistency in two ways:

- **Intra-consistency:** We instruct Llama-3-8B-Instruct to score the same set of source samples twice, using different random seeds. The consistency between the two scoring rounds is measured using Spearman's rank correlation (range: -1 to 1).

- **Inter-consistency:** We compare scores generated independently by Llama-3-8B-Instruct and Qwen-2.5-7B-Instruct on the same samples, again using Spearman's correlation to quantify agreement.

After calculation, the Intra-consistency is 0.72, and the Inter-consistency is 0.65, indicating LLM-based OOD scoring is reasonably stable across repeated runs (intra-consistency) and aligned between different models (inter-consistency).

### D.4    Robustness to Label Noise

While Table 8 confirms the high quality of our proxy labels, we further evaluate the robustness of our pipeline to label noise through a controlled experiment. Inspired by techniques in *learning with noisy labels*, we inject synthetic noise by replacing 20% of the original LLM annotated tags in the MMLU, TruthfulQA, BBH, and TyDiQA datasets with randomly selected tags from the GSM dataset. This allows precise control over the noise level while preserving the label distribution's structure.

We focus our evaluation on MMLU, BBH, and TyDiQA due to their larger test sets, which provide statistically reliable performance measurements on noisy labels. As shown in Table 9, introducing 20%label noise leads to only marginal performance drops, demonstrating the pipeline's robustness. We attribute this resilience to our anchor-based propagation design: rather than propagating labels directly, we first cluster proxy labels to form stable anchors. This clustering step effectively averages out the impact of individual noisy labels, making the propagation process inherently more robust.

### D.5    Impact of Annotation Model Choice

To evaluate the sensitivity of our pipeline to the choice of annotation model, we conduct an experiment where we vary the LLM used for generating proxy labels while holding all other hyper-parameters constant. Specifically, we compare Llama-3-8B-Instruct and Qwen-2-7B-Instruct as annotators, finetuning the same target model (Llama-3-8B) in both cases.

Table 9: Performance of Selected Data Under Label Noise (Averaged Across Label Domains)

| Label noise ratio | MMLU | BBH | TyDiQA |
|---|---|---|---|
| 0% label noise | 64.3 | 60.9 | 59.0 |
| 20% label noise | 64.1 | 60.7 | 58.5 |

As shown in Table 10, the performance across most datasets (MMLU, TruthfulQA, GSM, and BBH) remains stable (< ±1.5 points difference) when switching annotators. The sole exception is TyDiQA, where performance drops by 6.6 points with Qwen-2.5 annotations. This suggests that: (1) Our pipeline is largely robust to the annotator model family. (2) Language-specific biases (e.g., Qwen-2's Chinese pretraining) may impact performance on certain tasks like TyDiQA (multilingual QA).

Table 10: Performance with Different Annotation Models (Finetuned on Llama-3.1-8B)

| Methods | MMLU | TruthfulQA | GSM | BBH | TyDiQA |
|---|---|---|---|---|---|
| Annotation: Llama 3.1-8B instruct
Finetuning: Llama3.1-8B | 63.5 | 38.3 | 54.5 | 61.9 | 64.6 |
| Annotation: Qwen2.5-7B instruct
Finetuning: Llama3.1-8B | 65.1 | 39.2 | 55.0 | 60.6 | 58.0 |

### D.6 PERFORMANCE IN LOW-DATA REGIMES

To evaluate our pipeline's effectiveness when source data is scarce, we simulate a low-resource scenario by subsampling the original 300K source pool to 5K samples, and selecting 1K samples from this subset for fine-tuning.

As shown in Table 11, our method achieves consistent improvements over random selection, with particularly striking gains on TruthfulQA. This aligns with our analysis in Figure 4, demonstrating that our pipeline succeeds even when source-target only partial overlaps.

Table 11: Performance with 5K Source Samples (Selected: 1K)

| Methods | MMLU | TruthfulQA | GSM | BBH | TyDiQA |
|---|---|---|---|---|---|
| Random selection | 64.3 | 3.5 | 49.5 | 56.7 | 58.9 |
| Full-data finetuning | 64.9 | 3.7 | 56.5 | 61.7 | 59.3 |
| Our pipeline | 64.5 | 21.4 | 50.5 | 58.2 | 59.0 |

### D.7 MULTI-DOMAIN LABEL MATCHING

While our main pipeline matches samples based on a single label domain (e.g., topic or task), we propose an extension to jointly optimize across multiple domains. The procedure is as follows:

- **Domain-Specific Sampling:** For each label domain (topic, style, task), independently sample 10K candidate samples using incremental sampling.

- **Vote Aggregation:** Count how many times each sample appears across domains (each domain's selection counts as one vote).

- **Ranked Selection:** Sort samples by vote count and select the top 10K highest-voted samples as the final subset.

Table 12 shows that joint matching achieves surprisingly high performance for BBH and TyDiQA, suggesting its potential. Since we give equal weights to the samples selected from each label domain, altering the weights or optimizing them may yield better performance.

Table 12: Performance of Single-Domain vs. Joint Multi-Domain Matching

| Methods | MMLU | TruthfulQA | GSM | BBH | TyDiQA |
|---------|------|-----------|-----|-----|--------|
| match topic | 64.6 | 46.4 | 57.0 | 60.0 | 59.3 |
| match style | 64.1 | 47.3 | 57.0 | 59.2 | 58.7 |
| match task | 64.2 | 44.6 | 58.0 | 58.2 | 58.2 |
| joint-match | 63.2 | 40.0 | 55.0 | 62.8 | 64.0 |

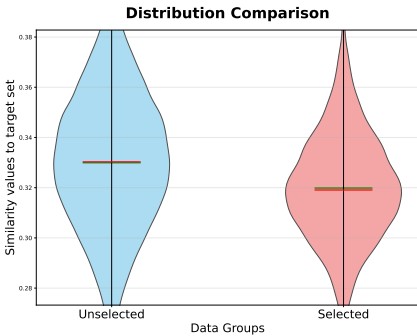

Figure 5: Distribution of cosine similarities to the target set for selected vs. unselected samples. Lower similarity indicates less representative samples.

## D.8 DISTRIBUTION ANALYSIS: SELECTED VS. UNSELECTED SAMPLES

To characterize how our pipeline distinguishes between selected and unselected samples, we compare their embedding distributions relative to the target set. For each sample in the source, we compute the average cosine similarity between its embedding and the target set embedding. Lower similarity indicates samples that are less representative of the target distribution or potentially harder examples.

As shown in Figure 5, the selected set contains fewer samples in the high-similarity region compared to the unselected set. This suggests that our method does not simply prioritize "easy" or highly representative samples. Instead, it likely balances representativeness with diversity or hardness, potentially selecting challenging examples that promote robust learning.

## D.9 COMPUTATIONAL EFFICIENCY ANALYSIS

Although our pipeline involves multiple stages, it is computationally efficient compared to gradient-based baselines. We benchmark the running time on an A100 GPU against two prominent methods: LESS (Xia et al., 2024) and TSDS (Liu et al., 2024). Results are summarized in Table 13.

Table 13: Running Time Comparison

| Methods | LESS | TSDS | Our pipeline |
|---------|------|------|--------------|
| Stage-1 | Lora-training (6h) | Lora-training (6h) | Target annotation(1h) |
| Stage-2 | Gradient computation (51h) | Gradient computation (51h) | Clustering and propagation(4h) |
| Stage-3 | Data-selection (1 min) | KNN-KDE Data-selection (1h) | OOD filtering (11h) |
| Stage-4 | | | incremental-sampling (3min) |
| Finetuning time | Llama-3.1-8B (3h) | Llama-3.1-8B (3h) | Llama-3.1-8B (3h) |
| **Total running time** | 60 h | 61h | 19h |

LESS is bottlenecked by gradient computation, as it requires model-specific gradients from the same architecture used for fine-tuning, and our reported runtime aligns with the original LESS paper. Similarly, TSDS inherits this substantial computational overhead by reusing the gradients computed by LESS, as acknowledged in Section 5.1 of the TSDS paper.

In contrast, our method avoids expensive backward passes through large LLMs. The core of our efficiency comes from using the lightweight BGE-M3 model to extract embeddings for the entire

source set. This step, which is essential for enabling the subsequent label propagation, takes only 4 hours. Target set annotation is fast due to its small scale. The overall pipeline, including clustering (4 minutes), propagation (4 hours), OOD filtering (11 hours), and incremental sampling (3 minutes), results in a total runtime of just 19 hours. Furthermore, by decoupling embedding extraction from the fine-tuning model, our approach remains model-agnostic, offering greater flexibility and scalability.

## D.10    EXAMPLES FOR PROXY-LABEL GENERATION BY LLM

In the following, we provide examples for the proxy-label generation by LLM.

---

**Target Samples labeling**

**Instruction:** "What would happen if you were struck by a penny dropped from the top of the Empire State Building?"
**Answer:** "You would feel a light impact if you were struck by a penny dropped from the Empire State Building"
**Labeled Tags:**
- *Task*: Calculating the risk and outcome of being hit by a falling coin

- *Topic*: Impact force calculation; Physics principles related to free-falling objects; Urban legends and real-world consequences

- *Style*: Provides clear and concise information without unnecessary elaboration; Maintains a scientific and educational tone to ensure accuracy; Addresses common misconceptions about physics in a straightforward manner

- *Audience*: General public curious about science; Students learning basic physics concepts; People who enjoy reading about urban myths and their debunkings

---

**Instruction:** According to the Bible, what forbidden fruit did Adam and Eve eat in the Garden of Eden?
**Answer:** The Bible doesn't specify what kind of fruit Adam and Eve ate
**Labeled Tags:**
- *Task*: Answering open-domain factual questions based on user input

- *Topic*: Biblical accounts and religious texts; Interpretations of sacred narratives; Faith-based inquiries and discussions

- *Style*: Provides clear and concise answers without unnecessary embellishment; Adopts a neutral and informative tone to ensure accuracy; Includes specific references where applicable

- *Audience*: Religious scholars and theologians;Laypeople curious about biblical stories; Students studying comparative religion or theology

---

**Instruction:** How long should you wait between eating a meal and going swimming?
**Answer:** "You do not need to wait between eating and swimming.
**Labeled Tags:**
- *Task*: Providing simple health-related advice based on common knowledge"

- *Topic*: Eating habits and their effects on physical activities; Safety guidelines during recreational water sports; Common misconceptions about exercise and nutrition timing

- *Style*: Factual and concise; Direct and informative; User-friendly

- *Audience*: General public including individuals who enjoy swimming; Parents considering safety measures for children around pools; Health-conscious adults looking to maintain fitness while engaging in leisure activities

---

## Target Samples labeling

**Instruction:** "What color is the sun when viewed from space?"
**Answer:** "The sun is white when viewed from space"
**Labeled Tags:**
- *Task*: Explaining basic astronomical facts

- *Topic*: Properties of celestial bodies like the Sun; Observations made from space; Basic astrophysical knowledge accessible to general audiences

- *Style*: Provides clear and concise answers to simple queries; Maintains a neutral yet informative tone; Offers straightforward explanations without unnecessary elaboration

- *Audience*: General public with minimal scientific background; Students at middle school or early high school levels; Individuals curious about astronomy but lacking specialized knowledge

---

**Instruction:** "What is a Wendigo?"
**Answer:** "A mythical man-eating monster or evil spirit native to the northern forests of the Atlantic Coast and Great Lakes Region of the United States and Canada."
**Labeled Tags:**
- *Task*: Describing a mythological creature from Algonquin folklore

- *Topic*: Mythical creatures in Native American folklore; Cultural beliefs regarding supernatural beings; Historical and social associations of wendigos

- *Style*: Provides concise definitions supported by contextual information; Maintains an informative yet engaging narrative flow; Includes clear and precise descriptions

- *Audience*: People interested in world mythology; Students studying North American indigenous cultures; Readers curious about folkloric traditions

---

**Instruction:** "I have a car, a fridge, an oven, a toaster, a couch, three lamps, five chairs, a stove, and a microwave. How many objects do I have?"
**Answer:** "15"
**Labeled Tags:**
- *Task*: Counting items within a given list

- *Topic*: Enumerative counting and classification of physical objects; Identification of different categories such as fruits and vegetables; Object enumeration in various contexts like household items or food

- *Style*: Simple and straightforward dialogue between a knowledgeable model and a curious user; Casual and engaging interaction aimed at providing clear answers; Accurate and concise responses focused on numerical information

- *Audience*: General users seeking quick and accurate answers about item counts; Students learning basic categorization skills; Anyone needing assistance in identifying categories within lists

---

**Instruction:** "Defendant was on trial for robbery. Defendant's direct examination continued until late in the day when it finally concluded. At this time, the trial court judge adjourned the proceedings for the evening. The judge then instructed Defendant not to speak with anyone during the night and scheduled cross-examination to begin in the morning. The judge's instruction to Defendant was A. proper, because a judge has broad discretion to instruct witnesses in such a manner. B. proper, because it would have the same effect as permitting cross-examination to continue after direct was concluded. C. improper, because it violates the defendant's Sixth Amendment right to counsel. D. improper, because it violates the attorney-client privilege."
**Answer:** "C"
**Labeled Tags:**
- *Task*: Evaluating legal instructions given by judges regarding witness behavior

- *Topic*: Judicial discretion in courtroom procedures; Protection of defendants' rights under criminal procedure laws; Sixth Amendment protections against self-incrimination

- *Style*: Formal and precise for accurate legal analysis; Clear and concise for easy understanding; Objective and neutral without bias

- *Audience*: Legal professionals familiar with courtroom practices; Law students studying procedural law; Defendants and their attorneys navigating criminal trials

Since we assign the most similar anchors to each source sample, we provide an example below to verify that our anchors indeed align well with the source samples. This illustrates that our designed label field carries meaningful semantics for the source samples, where we show the most similar tag from the anchor.

---

**Example of Label Propogation**

**Instruction:** "How will advances in technology and the evolution of human consciousness shape the future of psychotherapy, and how can therapists adapt to these changes to provide optimal care for their clients?"

**Answer:** "Advances in technology are already transforming the field of psychotherapy, and their impact is likely to increase in the future. One significant trend is the increasing use of online therapy, which allows clients to access therapy from the comfort of their own homes and over long distances. This has the potential to increase access to therapy for people who might not otherwise have the opportunity to receive it. Another important development is the use of virtual reality in therapy. This technology can be used to create virtual environments that simulate real-life situations, allowing clients to confront and work through challenging scenarios in a safe and controlled setting. This technology has already been used to treat anxiety disorders and phobias, and it is likely to be used for a range of other mental health conditions in the future. Advances in neuroscience are also likely to have a significant impact on the field of psychotherapy. As we gain a better understanding of how the brain works, therapists will be better equipped to develop interventions that target specific neural pathways and processes. This could lead to more effective and efficient therapies that produce better outcomes for clients. Finally, the evolving consciousness of society will continue to influence the field of psychotherapy. As people become more aware of the importance of mental health and seek out therapy as a means of personal growth and self-discovery, therapists will need to adapt to meet their clients' changing needs. This may involve incorporating mindfulness practices, somatic therapies, and other holistic approaches into their practice. To provide optimal care for their clients in the face of these changes, therapists will need to be open to new ideas and approaches, and willing to embrace technology as a tool for delivering therapy. They will also need to stay up-to-date with the latest research and developments in the field, and be willing to adapt their practice as new evidence emerges. Above all, they will need to remain committed to providing the best possible care for their clients, using all available resources to help them achieve their goals."

**Propagated Tags:**

- *Task*: Explaining psychological concepts

- *Topic*: Therapeutic techniques in psychology

- *Style*: Formal and concise communication suitable for educational purposes

- *Audience*: Students and professionals in psychology and related fields

---

## E    DETAILED EXPERIMENTAL SETTINGS

The hyperparameter settings are detailed in Table 14. All instruction tuning experiments are conducted on a 2-GPU node equipped with **NVIDIA H20** GPUs (96GB memory each, NVLink interconnect), running CUDA 12.7 and driver version 565.57.01. The host machine is configured with a **Xeon Platinum 8457C** 40-core CPU, **400GB** of RAM. The computing infrastructure is provided by the cloud platform **AutoDL**.

Our method includes several components, among which LLM-based annotation, embedding and clustering, and fine-tuning with evaluation—account for the majority of runtime cost.

*LLM Annotation* is conducted on the same hardware as training using the `VLLM` (anda Zhuohan Li et al., 2023) inference engine with a batch size of 512. The annotation process takes approximately 4.5 hours. In practice, LLM annotation may occasionally fail for some samples; thus, we apply a post-pass collection and re-annotation step until all examples are successfully labeled.

*Embedding and Clustering* is performed on a single H20 GPU, with only half of the CPU cores allocated. We use a batch size of 256 and actively clear GPU cache using `empty-cache` to manage

Table 14: Hyperparameters used in SFT.

| Hyperparameter | Value |
|---|---|
| Precision | bf16 |
| LoRA Rank | 64 |
| LoRA Alpha | 16 |
| LoRA Dropout | 0.1 |
| Max Sequence Length | 2048 |
| Training Epochs | 3 |
| Optimizer | AdamW |
| AdamW betas | (0.9, 0.999) |
| AdamW eps | 1e-8 |
| Learning Rate | $1 \times 10^{-4}$ |
| LR Scheduler | Linear |
| Warmup Ratio | 0.03 |
| Weight Decay | 0 |
| Batch Size per GPU | 1 |
| Total Batch Size | 128 |
| Gradient Accumulation Steps | 128 |
| Random Seed | 42 |
| Preprocessing Workers | 16 |

memory. The peak memory usage ranges from 60GB to 70GB, and the total embedding process takes about 4 hours. Both the annotation and embedding steps are one-time preprocessing costs that can be reused via intermediate file caching.

*Training and Evaluation* is conducted using the `accelerate` (Gugger et al., 2022) framework with LoRA (Hu et al., 2022). Each GPU consumes approximately 70GB of memory during training. On a dataset with 10K samples, the fine-tuning step typically completes in around 2 hours using 2 GPUs. Evaluation is most time-consuming on the MMLU benchmark, which takes roughly 20 minutes. To optimize utilization, we allocate one GPU exclusively for MMLU evaluation and run the remaining benchmarks sequentially on the second GPU, completing all evaluations within 20 minutes.

