# OpenReview forum: "Task-Aware Data Selection via Proxy-Label Enhanced Distribution Matching for LLM Finetuning"
_ICLR.cc/2026/Conference — ICLR 2026 Poster_

### Official Review · Reviewer_MYpb · 2025-10-24

**Soundness:** 2
**Presentation:** 2
**Contribution:** 2
**Rating:** 4
**Confidence:** 5

**Summary:**

This paper introduces a task-aware data selection pipeline for fine-tuning llms. The approach focuses on aligning both input features and task-specific labels to improve the relevance and quality of selected instruction data. Since task-specific labels are often unavailable, the method uses LLMs to generate proxy labels for the target dataset, which are clustered and propagated to the source dataset. Then a two-stage selection process first filters out low-quality examples using llm-based scoring, and then matches the label distribution through incremental selection.

**Strengths:**

- The paper extends prior work by considering additional alignment dimensions during task-specific data selection, including task, topic, style, and audience.
- It offers an information-theoretic explanation that provides a principled understanding of the proposed data selection method and prior works.

**Weaknesses:**

- The method relies heavily on LLM-based judgment, but does not evaluate the robustness or reliability. It remains unclear how accurate the generated labels are and how consistent or calibrated the LLM-assigned quality scores are.
- The approach introduces several hyper-parameters and control knobs (e.g., k in k-means clustering, minimum score thresholds, label alignment choices) without providing clear guidance on how to tune them. According to the experiments, the results are sensitive to them.
- The paper does not provide any theoretical guarantees for the proposed distribution alignment algorithm. It is unclear whether this algorithm will converge or output a better match.
- The experimental results lack error bars, making it difficult to assess statistical significance or robustness of the reported improvements.
- Minor presentation issues: in Figure 2, the text and numbers are too small and overlap, which affects readability.

**Questions:**

- Why you choose 100 as the number of centroids in k-means clustering?
- In label propagation, how exactly are the source examples embedded? Are they also labelled by the llm using the same process as the target examples?
- In "Prompt Template for Scoring Source Samples", the connection between the scoring instructions and the provided labels is unclear. What purpose do the labels serve in this context?
- Can we jointly match multiple labels using this method?

---

> ### Author Response · Authors · 2025-11-25
> **Part-1 of Our Reply**
>
> Thank you for your valuable time and insightful comment. We have carefully read your comments and have responded and explained accordingly.
>
>
>
>
> > **W1:** The method relies heavily on LLM-based judgment, but does not evaluate the robustness or reliability. It remains unclear how accurate the generated labels are and how consistent or calibrated the LLM-assigned quality scores are.
>
> **A:**  Thank you for raising this concern. We have conducted a thorough evaluation of the quality of proxy labels generated by the LLM, assessing both precision (whether a label correctly describes its corresponding instruction) and consistency (whether the same label is applied to semantically similar instructions). Our examining methodology follows the framework established by  [R1].
>
> - **For precision evaluation**: We randomly sampled 500 instruction-label pairs. GPT-4 was asked to assess if the label adequately describes the instruction. From these, the first 50 pairs were selected for human evaluation by three independent annotators.
> - **For consistency evaluation**: We randomly sampled 500 labels, ensuring each label was associated with at least two distinct instructions. GPT-4 was asked to determine if the instructions for a given label were semantically consistent. Similarly, the first 50 sets of instructions were assessed by three human annotators.
>
> We calculated the agreement between raters using Cohen’s Kappa (for pairwise agreement between human and GPT-4) and Fleiss’ Kappa (for inter-annotator agreement among humans). The results are presented below:
>
> | Metrics           | GPT-4 eval | Human-eval | Human-Human agreement | Human-GPT agreement |
> | ----------------- | ---------- | ---------- | --------------------- | ------------------- |
> | label precision   | 0.958      | 0.94       | 0.4823                | 0.7899              |
> | label consistency | 0.856      | 0.9        | 0.73                  | 0.736               |
>
> As shown, both GPT-4 and human evaluations indicate high precision and consistency for the proxy labels. The Human-GPT agreement scores (exceeding 0.7) indicate solid alignment between human and LLM judgments. We have uploaded the GPT and human evaluation scores, along with sampled instructions and labels, to our anonymized code repository: https://anonymous.4open.science/r/TADS-B14D/README.md.
>
>
>
> To further evaluate the robustness of our pipeline to label quality, we conducted a controlled noise injection experiment. Inspired by methodologies in "Learning with Noisy Labels," we inject synthetic noise by replacing 20% of the original LLM annotated tags in the MMLU, TruthfulQA, BBH, and TyDiQA datasets with randomly selected tags from the GSM dataset. This allows precise control over the noise level while preserving the label distribution’s structure.
>
> We focus our evaluation on MMLU, BBH, and TyDiQA due to their larger test sets, which provide statistically reliable performance measurements on noisy labels. As shown in Table below, introducing 20%label noise leads to only marginal performance drops, demonstrating the pipeline’s robustness. We attribute this resilience to our anchor-based propagation design: rather than propagating labels directly, we first cluster proxy labels to form stable anchors. This clustering step effectively averages out the impact of individual noisy labels, making the propagation process inherently more robust.
>
> |                 | MMLU | BBH  | TyDiQA |
> | --------------- | ---- | ---- | ------ |
> | No noise        | 64.3 | 60.9 | 59.0   |
> | 20% label noise | 64.1 | 60.7 | 58.5   |
>
>
>
> To assess the reliability of LLM-based OOD filtering, we evaluate score consistency in two ways:
>
> - Intra-consistency: We instruct Llama-3-8B-Instruct to score the same set of source samples twice, using different random seeds. The consistency between the two scoring rounds is measured using Spearman’s rank correlation (range: -1 to 1).
> - Inter-consistency: We compare scores generated independently by Llama-3-8B-Instruct and Qwen-2.5-7B-Instruct on the same samples, again using Spearman’s correlation to quantify agreement.
>
> After calculation, the Intra-consistency is 0.72, and the Inter-consistency is 0.65, indicating LLM-based OOD scoring is reasonably stable across repeated runs (intra-consistency) and aligned between different models (inter-consistency).
>
> We have added these analyses in Appendix D.2, Appendix D.3 and Appendix D.4.
>
> [R1] \#InsTag: Instruciton Tagging for Analyzing Supervised Fine-tuning of Large Language Models. ICLR 2024

---

> ### Author Response · Authors · 2025-11-25
> **Part-2 of Our Reply**
>
> > **W2:** The approach introduces several hyper-parameters and control knobs (e.g., k in k-means clustering, minimum score thresholds, label alignment choices) without providing clear guidance on how to tune them. According to the experiments, the results are sensitive to them.
>
> **A:**  We have conducted ablation studies regarding 'k' in Figure 3 in our main paper. The results show that our method is robust to 'k' when it ranges from 50 to 150.
>
> Our use of four label domains (task, topic, style, audience) primarily serves to highlight the flexibility of our framework. We apply these **general** label domains to different datasets. In practice, if we have little knowledge of the task, we can apply general label domains. However, if we have prior knowledge of the task, the choice of label domain can be guided by this knowledge. For example, in mathematical reasoning tasks, one could define the label domain as "mathematical problem-solving skills," as seen in [R1], which shows promise in improving exemplar quality using domain-specific labels. A more **explicit** domain definition can lead to more accurate data selection.
>
> As shown in Table 3 in our main paper, when applying **general** label domains ( task, topic, and style) to the five datasets, performance remains relatively stable when the minimum score exceeds 7 and our pipeline have strong performance on MMLU, TruthfulQA and GSM datasets compared to other task-specific data selection methods (LESS and TSDS).
>
> [R1] Metacognitive Capabilities of LLMs: An Exploration in Mathematical Problem Solving. NeurlPS 2024
>
>
>
> > **W3:** The paper does not provide any theoretical guarantees for the proposed distribution alignment algorithm. It is unclear whether this algorithm will converge or output a better match.
>
> **A:**  Thank you for raising this point. The primary contribution of our work is **a reconceptualization of the task-specific data selection problem for instruction tuning (Section 3)**, proposing joint alignment instead of input-only alignment. Our current pipeline serves as an initial instantiation of this approach. Through our experiments, we have demonstrated its effectiveness in task-specific data selection. We agree that theoretical guarantees are an important direction for future work and we will focus on developing theoretical guarantees moving forward.
>
>
>
> > **W4:** The experimental results lack error bars, making it difficult to assess statistical significance or robustness of the reported improvements.
>
> **A:**  We appreciate the reviewer's comment regarding the need for error bars to establish statistical significance. To provide this, we have conducted multiple runs (n=3) for the baseline methods and report the mean ± standard deviation in the table below. The analysis reveals that our primary methods ("align topic," "align style," and "align task") consistently show low variance (with no standard deviation exceeding 1.0), indicating stable and reliable improvements. Conversely, higher variance is observed for "Random Selection" and, somewhat, for "align audience. This suggests that the "audience label" domain may be less stable. Due to time constraints, we will run the remaining baselines and update all results in the final version of the paper.
>
>
>
> |                                    | MMLU         | TruthfulQA   | GSM          | BBH          | TyDiQA       |
> | ---------------------------------- | ------------ | ------------ | ------------ | ------------ | ------------ |
> | Random Selection                   | 63.70 ± 0.24 | 29.07 ± 3.41 | 54.00 ± 1.22 | 60.20 ± 0.33 | 60.50 ± 0.33 |
> | LESS                               | 63.25 ± 0.04 | 35.13 ± 1.41 | 56.05 ± 0.61 | 61.05 ± 0.54 | 64.67 ± 0.54 |
> | TSDS                               | 63.60 ± 0.08 | 44.05 ± 0.69 | 50.00 ± 1.22 | 62.05 ± 0.22 | 63.45 ± 0.54 |
> | Ours (align topic; min score 7)    | 64.45 ± 0.12 | 46.85 ± 0.29 | 56.95 ± 0.22 | 60.15 ± 0.29 | 58.55 ± 0.61 |
> | Ours (align style; min score 7)    | 64.20 ± 0.08 | 47.15 ± 0.12 | 57.75 ± 0.61 | 58.85 ± 0.29 | 59.25 ± 0.45 |
> | Ours (align task; min score 7)     | 64.40 ± 0.16 | 46.45 ± 0.53 | 57.50 ± 0.41 | 58.90 ± 0.57 | 59.30 ± 0.90 |
> | Ours (align audience; min score 7) | 64.35 ± 0.21 | 47.10 ± 1.22 | 55.75 ± 1.13 | 58.50 ± 0.73 | 60.20 ± 1.05 |
>
>
>
> > **W5:** Minor presentation issues: in Figure 2, the text and numbers are too small and overlap, which affects readability.
>
> **A:**  Thank you for your suggestion. We have revised Figure 2 to improve readability.
>
>
>
> > **Q1:** Why you choose 100 as the number of centroids in k-means clustering?
>
> **A:**  We added ablation studies on 'k' in Figure 3 in our main paper, which show that our pipeline is robust to 'k' when it ranges from 50 to 150.

---

> ### Author Response · Authors · 2025-11-25
> **Part-3 of Our Reply**
>
> > **Q2:** In label propagation, how exactly are the source examples embedded? Are they also labelled by the llm using the same process as the target examples?
>
> **A:**  The labels of source samples are derived through a propagation process designed for robustness. We first cluster the target domain's LLM-generated proxy labels into semantic anchors. Then, we compute the BGE-m3 embeddings for both the source examples and these anchors. Each source example is assigned the label of the anchor with which it has the highest cosine similarity. This method of using cluster-based anchors, rather than propagating individual and potentially noisy proxy labels, provides greater tolerance to label noise (see Appendix D.4).
>
> > **Q3:** In "Prompt Template for Scoring Source Samples", the connection between the scoring instructions and the provided labels is unclear. What purpose do the labels serve in this context?
>
> **A:**  This step leverages the LLM to evaluate the likelihood that a source text instruction belongs to its newly assigned label from the propagation step. This is critical for identifying open-set label noise, where irrelevant instructions are assigned to an target label. The LLM's score for a given instruction-label pair acts as a proxy for label noise detection; a lower score indicates a higher probability of the sample being noisy and a candidate for filtration. This method provides a direct way to filter noise without training a separate model, as discussed in Appendix C.
>
>
>
> > **Q4:** Can we jointly match multiple labels using this method?
>
> **A:**  Thank you for your very insightful comment! Following your comment, we propose an extension to jointly optimize across multiple domains. The procedure is as follows:
>
> - For each label domain (topic, style, task), independently sample 10K candidate samples using incremental sampling.
>
> - Count how many times each sample appears across domains (each domain’s selection counts as one vote).
>
> - Sort samples by vote count and select the top 10K highest-voted samples as the final subset.
>
> Table below shows that joint matching achieves surprisingly high performance for BBH and TyDiQA, suggesting its potential. Since we give equal weights to the samples selected from each label domain,
> altering the weights or optimizing them may yield better performance. We have added the code for joint matching in our anonymized repository: https://anonymous.4open.science/r/TADS-B14D/README.md and add this part of discussion in Appendix D.7.
>
> |             | MMLU | TruthfulQA | GSM  | BBH  | TyDiQA |
>   | ----------- | ---- | ---------- | ---- | ---- | ------ |
>   | match topic | 64.6 | 46.4       | 57.0 | 60.0 | 59.3   |
>   | match style | 64.1 | 47.3       | 57.0 | 59.2 | 58.7   |
>   | match task  | 64.2 | 44.6       | 58.0 | 58.2 | 58.2   |
>   | joint-match | 63.2 | 40.2       | 55.0 | 62.8 | 64.0   |

---

> ### Comment · Reviewer_MYpb · 2025-11-25
>
> Thank you for the detailed response. The clarifications addressed most of my concerns, and I will increase my score accordingly.

---

> > ### Author Response · Authors · 2025-11-26
> >
> > Thank you for your valuable feedback and active engagement during the rebuttal process. We are grateful for your endorsement!

---

### Official Review · Reviewer_CwHM · 2025-10-26

**Soundness:** 3
**Presentation:** 3
**Contribution:** 3
**Rating:** 6
**Confidence:** 4

**Summary:**

The paper tackles the problem of selecting high‐quality, task‐relevant instruction data for fine-tuning LLMs. The authors argue that existing data-selection methods focus only on aligning the input distribution X (i.e., instructions) with a target task, but neglect the joint distribution of (X,Y) where Y are task‐specific labels, which are often unavailable in practice. They propose a pipeline that uses an LLM to infer proxy labels for a large unlabeled source corpus, then applies a proxy-label enhanced distribution matching method: first filtering out noisy out-of-distribution samples, then aligning the remaining data to the target joint distribution (X,Y), and finally selecting a subset. Experiments show that fine-tuning on the selected subset can achieve performance competitive with or superior to using full dataset, thereby demonstrating task‐aware data selection is effective.

**Strengths:**

Novel viewpoint: Using proxy labels and distribution matching for task‐aware rather than input‐only data selection is an interesting insight.

Practical relevance: Demonstrating that a small subset of data can yield competitive fine‐tuning results addresses the real challenge of data efficiency in LLM tuning.

Clear presentation of the pipeline and motivation, making the method relatively easy to understand and adopt.

**Weaknesses:**

Proxy labels may introduce noise, and the paper gives limited analysis of how label quality affects downstream performance.

Transparency of cost/efficiency: While the claim of “smaller subset yields full‐data performance” is compelling, more detailed breakdowns (hardware, runtime, selection cost) would improve trust.

Risk of selection bias: Since the method selects based on proxy‐label generated metrics and distribution matching, it may inadvertently favour certain types of samples (e.g., easier ones, more model‐familiar) and perhaps neglect rare or hard tasks; the paper does not deeply analyse this risk.

Engineering complexity & scalability: Generating proxy labels, filtering, and distribution matching add overhead; discussion of how this scales or works in resource‐limited environments is limited.

**Questions:**

Can you report detailed metrics on proxy label quality: e.g., accuracy, noise rate, and how selection performance degrades (or improves) with differing label quality?

How robust is the method to different model architectures or sizes? If the fine-tune target model is quite different (size, family) from the one used to infer proxy labels, how does performance change?

Could you provide full cost breakdowns (selection cost + fine-tune cost + hardware) for your method and key baselines (input‐only selection, random sampling) under identical hardware?

Have you analysed the selected subset in terms of diversity: task types, difficulty levels, rare vs common categories, language styles? Is there any systematic bias in what gets selected vs discarded?

In truly low-data regimes (e.g., 1 K or 5 K samples) or for very niche tasks (domain‐specific), how does your method perform relative to full‐data or random sampling?

---

> ### Author Response · Authors · 2025-11-25
> **Part-1 of Our Reply**
>
> Thank you for your valuable time and insightful comment. We have carefully read your comments and have responded and explained accordingly.
>
> > **W1:** Proxy labels may introduce noise, and the paper gives limited analysis of how label quality affects downstream performance.
>
> **A:**  Thank you for raising this concern. We have conducted a thorough evaluation of the quality of proxy labels generated by the LLM, assessing both precision (whether a label correctly describes its corresponding instruction) and consistency (whether the same label is applied to semantically similar instructions). Our examining methodology follows the framework established by  [R1].
>
> - **For precision evaluation**: We randomly sampled 500 instruction-label pairs. GPT-4 was asked to assess if the label adequately describes the instruction. From these, the first 50 pairs were selected for human evaluation by three independent annotators.
> - **For consistency evaluation**: We randomly sampled 500 labels, ensuring each label was associated with at least two distinct instructions. GPT-4 was asked to determine if the instructions for a given label were semantically consistent. Similarly, the first 50 sets of instructions were assessed by three human annotators.
>
> We calculated the agreement between raters using Cohen’s Kappa (for pairwise agreement between human and GPT-4) and Fleiss’ Kappa (for inter-annotator agreement among humans). The results are presented below:
>
> | Metrics           | GPT-4 eval | Human-eval | Human-Human agreement | Human-GPT agreement |
> | ----------------- | ---------- | ---------- | --------------------- | ------------------- |
> | label precision   | 0.958      | 0.94       | 0.4823                | 0.7899              |
> | label consistency | 0.856      | 0.9        | 0.73                  | 0.736               |
>
> As shown, both GPT-4 and human evaluations indicate high precision and consistency for the proxy labels. The Human-GPT agreement scores (exceeding 0.7) indicate solid alignment between human and LLM judgments. We have uploaded the GPT and human evaluation scores, along with sampled instructions and labels, to our anonymized code repository: https://anonymous.4open.science/r/TADS-B14D/README.md.
>
>
>
> To further evaluate the robustness of our pipeline to label quality, we conducted a controlled noise injection experiment. Inspired by methodologies in "Learning with Noisy Labels," we inject synthetic noise by replacing 20% of the original LLM annotated tags in the MMLU, TruthfulQA, BBH, and TyDiQA datasets with randomly selected tags from the GSM dataset. This allows precise control over the noise level while preserving the label distribution’s structure.
>
> We focus our evaluation on MMLU, BBH, and TyDiQA due to their larger test sets, which provide statistically reliable performance measurements on noisy labels. As shown in Table below, introducing 20% label noise leads to only marginal performance drops, demonstrating the pipeline’s robustness. We attribute this resilience to our anchor-based propagation design: rather than propagating labels directly, we first cluster proxy labels to form stable anchors. This clustering step effectively averages out the impact of individual noisy labels, making the propagation process inherently more robust.
>
> |                 | MMLU | BBH  | TyDiQA |
> | --------------- | ---- | ---- | ------ |
> | No noise        | 64.3 | 60.9 | 59.0   |
> | 20% label noise | 64.1 | 60.7 | 58.5   |
>
> We have added these analyses in Appendix D.2 and Appendix D.4.
>
> [R1] \#InsTag: Instruciton Tagging for Analyzing Supervised Fine-tuning of Large Language Models. ICLR 2024

---

> ### Author Response · Authors · 2025-11-25
> **Part-2 of Our Reply**
>
> > **W2:** Transparency of cost/efficiency: While the claim of “smaller subset yields full‐data performance” is compelling, more detailed breakdowns (hardware, runtime, selection cost) would improve trust.
>
> **A:** Thank you for your comment. Although our pipeline involves multiple stages, it is computationally efficient compared to gradient-based baselines. We benchmark the running time on an A100 GPU  against two prominent methods: LESS and TSDS. Results are summarized below:
>
>
>
> |      | Stage-1                    | Stage-2                                   | Stage-3                     | Stage-4                     | Finetuning        | Total-time |
> | ---- | -------------------------- | ----------------------------------------- | --------------------------- | --------------------------- | ----------------- | ---------- |
> | LESS | Lora-training (6h)         | Gradient computation (51h)                | Data-selection (1 min)      |                             | Llama-3.1 8B (3h) | 60 h       |
> | TSDS | Lora-training (6h)         | Gradient computation (51h)                | KNN-KDE Data-selection (1h) |                             | Llama-3.1 8B (3h) | 61h        |
> | Ours | Target set annotation (1h) | Clustering and propagation to source (4h) | OOD filtering (11h)         | incremental-sampling (3min) | Llama-3.1 8B (3h) | 19h        |
>
>
>
> LESS is bottlenecked by gradient computation, as it requires model-specific gradients from the same architecture used for fine-tuning, and our reported runtime aligns with the original LESS paper. Similarly, TSDS inherits this substantial computational overhead by reusing the gradients computed by LESS, as acknowledged in Section 5.1 of the TSDS paper.
>
> In contrast, our method avoids expensive backward passes through large LLMs. The core of our efficiency comes from using the lightweight BGE-M3 model to extract embeddings for the entire source set. This step, which is essential for enabling the subsequent label propagation, takes only 4 hours. Target set annotation is fast due to its small scale. The overall pipeline, including clustering (4 minutes), propagation (4 hours), OOD filtering (11 hours), and incremental sampling (3 minutes), results in a total runtime of just 19 hours. Furthermore, by decoupling embedding extraction from the fine-tuning model, our approach remains model-agnostic, offering greater flexibility and scalability.
>
> This running-time analysis has been included in Appendix D.9 in our paper.
>
>
>
> > **W3:** Risk of selection bias: Since the method selects based on proxy‐label generated metrics and distribution matching, it may inadvertently favour certain types of samples (e.g., easier ones, more model‐familiar) and perhaps neglect rare or hard tasks; the paper does not deeply analyse this risk.
>
>
>
> **A:** Thank you for your comment. We agree that LLM-based data selection may introduce bias, which is indeed a common challenge in methods of this kind (e.g., task-unspecific selection approaches [R1, R2]).  In our pipeline, potential bias could arise from two main sources: LLM annotation and LLM scoring. In response to your suggestion, we have quantitatively assessed these aspects.
>
> - **LLM Annotation Consistency:** As mentioned in **W1**, the precision and consistency between LLM annotations and human evaluations are over 0.7, suggesting solid agreement.
>
> - **LLM Scoring and Sampling Strategy:**  After the LLM assigns quality scores to samples based on their assigned anchors, a naive approach of selecting only high-scoring samples could indeed bias the fine-tuning process toward easier examples, potentially overlooking rare or challenging instances. To mitigate this, we employ incremental sampling designed to align the selected subset with the target distribution rather than simply maximizing scores.
>
>   To further validate this approach, we analyzed the embedding distributions of selected versus unselected samples relative to the target set. For each source sample, we computed the average cosine similarity to embeddings of the target set, where lower similarity indicates less representative or potentially harder examples. As shown in Figure 5 (Appendix D.8), the selected set contains fewer samples in the high-similarity region compared to the unselected set. This suggests that our method does not simply prioritize "easy" or highly aligned samples but instead balances representativeness with diversity, likely including challenging examples that support more robust model learning.
>
> In future work, we will continue working to minimize bias while maintaining strong performance.
>
> [R1] AlpaGasus: Training A Better Alpaca with Fewer Data. ICLR 2024
>
> [R2] What Makes Good Data for Alignment? A Comprehensive Study of Automatic Data Selection in Instruction Tuning. ICLR 2024

---

> ### Author Response · Authors · 2025-11-25
> **Part-3 of Our Reply**
>
> > **W4:** Engineering complexity & scalability: Generating proxy labels, filtering, and distribution matching add overhead; discussion of how this scales or works in resource‐limited environments is limited.
>
> **A:** Despite our pipeline having four stages, none of them require LLM fine-tuning. Refer to our response to **W2**, where we show that the most resource-intensive steps is OOD sampling. These stages scale linearly with the number of query examples and source examples. Unlike LESS and TSDS, which require costly gradient computations by LLM, we use BGE-m3 for embedding extraction, which is much smaller and more resource-efficient, making it suitable for resource-limited environments.
>
>
>
> > **Q1:** Can you report detailed metrics on proxy label quality: e.g., accuracy, noise rate, and how selection performance degrades (or improves) with differing label quality?
>
> **A:** Please refer to our response to **W1.**
>
>
> > **Q2:** How robust is the method to different model architectures or sizes? If the fine-tune target model is quite different (size, family) from the one used to infer proxy labels, how does performance change?
>
> **A:** To evaluate the sensitivity of our pipeline to the choice of annotation model, we conduct an experiment where we vary the LLM used for generating proxy labels while holding all other hyper-parameters constant. Specifically, we compare Llama-3-8B-Instruct and Qwen-2.5-7B-Instruct as annotators, finetuning the same target model (Llama-3-8B) in both cases.
>
> As shown in Table below, the performance across most datasets (MMLU, TruthfulQA, GSM, and BBH) remains stable (< ±1.5 points difference) when switching annotators. The sole exception is TyDiQA, where performance drops by 6.6 points with Qwen-2.5 annotations. This suggests that: (1) Our pipeline is largely robust to the annotator model family. (2) Language-specific biases (e.g., Qwen-2’s Chinese pretraining) may impact performance on certain tasks like TyDiQA (multilingual QA).
>
>
>
> | Setting                                                      | MMLU | TruthfulQA | GSM  | BBH  | TydiQA |
> | ------------------------------------------------------------ | ---- | ---------- | ---- | ---- | ------ |
> | Annotation: Llama 3.1 8B instruct;  Finetuning model : Llama3.1 8B | 63.5 | 38.3       | 54.5 | 61.9 | 64.6   |
> | Annotation: Qwen2.5-7B instruct;  Finetuning model : Llama3.1  8B | 65.1 | 39.2       | 55.0 | 60.6 | 58.0   |
>
> This part of analyses has been added to Appendix D.5.
>
>
>
> > **Q3:** Could you provide full cost breakdowns (selection cost + fine-tune cost + hardware) for your method and key baselines (input‐only selection, random sampling) under identical hardware?
>
> **A:** Please refer to our response to **W2**.
>
>
>
> > **Q4:** Have you analysed the selected subset in terms of diversity: task types, difficulty levels, rare vs common categories, language styles? Is there any systematic bias in what gets selected vs discarded?
>
> **A:** Please refer to our response to **W3**.
>
>
>
> > **Q5:** In truly low-data regimes (e.g., 1 K or 5 K samples) or for very niche tasks (domain‐specific), how does your method perform relative to full‐data or random sampling?
>
> **A:**  Thank you for your question. To evaluate our pipeline’s effectiveness when source data is scarce, we simulate a low-resource scenario by subsampling the original 300K source pool to 5K samples, and selecting 1K samples from this subset for fine-tuning.
>
>
>
> As shown in Table below, our method achieves consistent improvements over random selection, with particularly striking gains on TruthfulQA. This aligns with our analysis in Figure 4, demonstrating that our pipeline succeeds even when source-target only partial overlaps.
>
>
>
> | Methods              | MMLU | TruthfulQA | GSM  | BBH  | TydiQA |
> | -------------------- | ---- | ---------- | ---- | ---- | ------ |
> | Random selection     | 64.3 | 3.5        | 49.5 | 56.7 | 58.9   |
> | Full-data finetuning | 64.9 | 3.5        | 56.5 | 61.7 | 59.3   |
> | Our pipeline         | 64.5 | 21.4       | 50.5 | 58.2 | 59.0   |
>
> This part of analyses has been added to Appendix D.6.

---

### Official Review · Reviewer_dz9h · 2025-10-30

**Soundness:** 2
**Presentation:** 2
**Contribution:** 2
**Rating:** 6
**Confidence:** 1

**Summary:**

This paper introduces a proxy-label enhanced joint distribution matching approach for task-specific data selection in large language model fine-tuning. The key idea is to let the model generate task-related proxy labels so that both inputs and outputs are considered jointly when aligning distributions, rather than focusing only on input similarity.

As a researcher specializing in human-computer interaction, this study clearly falls outside my area of expertise. Following the Area Chair’s instructions, I have selected “1: You are unable to assess this paper and have alerted the ACs to seek an opinion from different reviewers” and submitted my review accordingly. Therefore, I will not be participating in the rebuttal stage for this manuscript. Thank you for your understanding.

**Strengths:**

1. The paper reformulates task-specific data selection as a joint distribution alignment problem, moving beyond traditional input-only approaches. The introduction of proxy labels adds a fresh perspective to modeling task relevance.

2. It proposes a complete and coherent pipeline, from proxy-label generation and clustering to noise filtering and incremental sampling, with clear logical flow and information-theoretic grounding.

3. Experimental results on multiple mainstream benchmarks, such as MMLU, TruthfulQA, and GSM8K, show stable or superior performance compared with SOTA methods like LESS and TSDS, especially under low-data conditions.

**Weaknesses:**

1. Although the paper uses LLMs to generate proxy labels, the analysis of their consistency, bias, and noise propagation is rather superficial and lacks quantitative evaluation or comparison with human annotations.

2. The multi-stage pipeline, involving annotation, clustering, filtering, and sampling, lacks detailed efficiency analysis on large-scale corpora. Its scalability and real-world deployment cost remain unclear.

3. The experiments are limited to English and general-purpose LLMs like LLaMA and Mistral. There is little discussion on adaptation to multilingual or multimodal tasks, and the explanation for performance drop on TyDiQA is vague.

**Questions:**

1. Can the authors provide an evaluation of the consistency or confidence of LLM-generated proxy labels compared with human annotations to verify label quality?

2. Can the proposed method generalize to cross-domain or cross-lingual scenarios, such as transferring from legal to medical tasks? Would new proxy labels be required in such cases?

3. Are the hyperparameters and target set sizes for LESS and TSDS exactly matched to those used in this paper? If not, this should be clearly stated to ensure fair comparison.

4. Have the authors analyzed the sensitivity of key parameters, such as the minimum score threshold or the number of clusters k? Without this, reproducibility and transferability could be limited.

5. The ablation only examines the combined effect of filtering and sampling. It would be helpful to further analyze how each stage contributes to different task types, such as reasoning, factual, or comprehension tasks.

---

> ### Author Response · Authors · 2025-11-25
> **Part-1 of Our Reply**
>
> Thank you for your valuable time and insightful comment. We have carefully read your comments and have responded and explained accordingly.
>
> > **W1:** Although the paper uses LLMs to generate proxy labels, the analysis of their consistency, bias, and noise propagation is rather superficial and lacks quantitative evaluation or comparison with human annotations.
>
> **A:**  Thank you for raising this concern. We have conducted a thorough evaluation of the quality of proxy labels generated by the LLM, assessing both precision (whether a label correctly describes its corresponding instruction) and consistency (whether the same label is applied to semantically similar instructions). Our examining methodology follows the framework established by  [R1].
>
> - **For precision evaluation**: We randomly sampled 500 instruction-label pairs. GPT-4 was asked to assess if the label adequately describes the instruction. From these, the first 50 pairs were selected for human evaluation by three independent annotators.
> - **For consistency evaluation**: We randomly sampled 500 labels, ensuring each label was associated with at least two distinct instructions. GPT-4 was asked to determine if the instructions for a given label were semantically consistent. Similarly, the first 50 sets of instructions were assessed by three human annotators.
>
> We calculated the agreement between raters using Cohen’s Kappa (for pairwise agreement between human and GPT-4) and Fleiss’ Kappa (for inter-annotator agreement among humans). The results are presented below:
>
> | Metrics           | GPT-4 eval | Human-eval | Human-Human agreement | Human-GPT agreement |
> | ----------------- | ---------- | ---------- | --------------------- | ------------------- |
> | label precision   | 0.958      | 0.94       | 0.4823                | 0.7899              |
> | label consistency | 0.856      | 0.9        | 0.73                  | 0.736               |
>
> As shown, both GPT-4 and human evaluations indicate high precision and consistency for the proxy labels. The Human-GPT agreement scores (exceeding 0.7) indicate solid alignment between human and LLM judgments. We have uploaded the GPT and human evaluation scores, along with sampled instructions and labels, to our anonymized code repository: https://anonymous.4open.science/r/TADS-B14D/README.md.
>
> To further evaluate the robustness of our pipeline to label quality, we conducted a controlled noise injection experiment. Inspired by methodologies in "Learning with Noisy Labels," we inject synthetic noise by replacing 20% of the original LLM annotated tags in the MMLU, TruthfulQA, BBH, and TyDiQA datasets with randomly selected tags from the GSM dataset. This allows precise control over the noise level while preserving the label distribution’s structure.
>
> We focus our evaluation on MMLU, BBH, and TyDiQA due to their larger test sets, which provide statistically reliable performance measurements on noisy labels. As shown in Table below, introducing 20% label noise leads to only marginal performance drops, demonstrating the pipeline’s robustness. We attribute this resilience to our anchor-based propagation design: rather than propagating labels directly, we first cluster proxy labels to form stable anchors. This clustering step effectively averages out the impact of individual noisy labels, making the propagation process inherently more robust.
>
> |                 | MMLU | BBH  | TyDiQA |
> | --------------- | ---- | ---- | ------ |
> | No noise        | 64.3 | 60.9 | 59.0   |
> | 20% label noise | 64.1 | 60.7 | 58.5   |
>
> We have added these analyses in Appendix D.2 and Appendix D.4.
>
> [R1] \#InsTag: Instruciton Tagging for Analyzing Supervised Fine-tuning of Large Language Models. ICLR 2024

---

> ### Author Response · Authors · 2025-11-25
> **Part-2 of Our Reply**
>
> > **W2:** The multi-stage pipeline, involving annotation, clustering, filtering, and sampling, lacks detailed efficiency analysis on large-scale corpora. Its scalability and real-world deployment cost remain unclear.
>
> **A:** Thank you for your comment. Although our pipeline involves multiple stages, it is computationally efficient compared to gradient-based baselines. We benchmark the running time on an A100 GPU  against two prominent methods: LESS and TSDS. Results are summarized below:
>
> |      | Stage-1                    | Stage-2                                   | Stage-3                     | Stage-4                     | Finetuning        | Total-time |
> | ---- | -------------------------- | ----------------------------------------- | --------------------------- | --------------------------- | ----------------- | ---------- |
> | LESS | Lora-training (6h)         | Gradient computation (51h)                | Data-selection (1 min)      |                             | Llama-3.1 8B (3h) | 60 h       |
> | TSDS | Lora-training (6h)         | Gradient computation (51h)                | KNN-KDE Data-selection (1h) |                             | Llama-3.1 8B (3h) | 61h        |
> | Ours | Target set annotation (1h) | Clustering and propagation to source (4h) | OOD filtering (11h)         | incremental-sampling (3min) | Llama-3.1 8B (3h) | 19h        |
>
>
>
> LESS is bottlenecked by gradient computation, as it requires model-specific gradients from the same architecture used for fine-tuning, and our reported runtime aligns with the original LESS paper. Similarly, TSDS inherits this substantial computational overhead by reusing the gradients computed by LESS, as acknowledged in Section 5.1 of the TSDS paper.
>
> In contrast, our method avoids expensive backward passes through large LLMs. The core of our efficiency comes from using the lightweight BGE-M3 model to extract embeddings for the entire source set. This step, which is essential for enabling the subsequent label propagation, takes only 4 hours. Target set annotation is fast due to its small scale. The overall pipeline, including clustering (4 minutes), propagation (4 hours), OOD filtering (11 hours), and incremental sampling (3 minutes), results in a total runtime of just 19 hours. Furthermore, by decoupling embedding extraction from the fine-tuning model, our approach remains model-agnostic, offering greater flexibility and scalability.
>
> This running-time analysis has been included in Appendix D.9 in our paper.
>
>
>
> > **W3:** The experiments are limited to English and general-purpose LLMs like LLaMA and Mistral. There is little discussion on adaptation to multilingual or multimodal tasks, and the explanation for performance drop on TyDiQA is vague.
>
>
>
> **A:** Our task-specific data selection experiments were designed following related works (LESS and TSDS) and using the same model family (LLaMA and Mistral). Compared to LESS and TSDS, we add more datasets to cover diverse real-world tasks.
>
> The performance drop on TyDiQA could be due to the following factors: 1. TyDiQA is a multilingual dataset, and the 8B model may not have sufficient capacity to understand multilingual semantics. 2. The single label fields we designed (Task, Topic, Style, Audience) may not fully capture TyDiQA’s inherent characteristics. However, in Appendix D.7, we find that align multiple-label domain can achieve very good performance on TyDiQA.
>
>
> > **Q1:** Can the authors provide an evaluation of the consistency or confidence of LLM-generated proxy labels compared with human annotations to verify label quality?
>
> **A:** Please refer to our response to **W1**.
>
> > **Q2:** Can the proposed method generalize to cross-domain or cross-lingual scenarios, such as transferring from legal to medical tasks? Would new proxy labels be required in such cases?
>
> **A:** If domain gap is not large, then we may not need to let LLM re-annotate the labels. From Appendix D.4, our pipline is robust to certain label noise. However, However, if there is a notable domain gap, such as transferring from legal to medical tasks, new proxy labels will indeed be required.
>
> Our use of four label domains (task, topic, style, audience) primarily serves to highlight the flexibility of our framework. We apply these **general** label domains to different datasets. In practice, if we have little knowledge of the task, we can apply general label domains. However, if we have prior knowledge of the task, the choice of label domain can be guided by this knowledge. For example, in mathematical reasoning tasks, one could define the label domain as "mathematical problem-solving skills," as seen in [R1], which shows promise in improving exemplar quality using domain-specific labels. A more **explicit** domain definition can lead to more accurate data selection.
>
> [R1] Metacognitive Capabilities of LLMs: An Exploration in Mathematical Problem Solving. NeurlPS 2024

---

> ### Author Response · Authors · 2025-11-25
> **Part-3 of Our Reply**
>
> > **Q3:** Are the hyperparameters and target set sizes for LESS and TSDS exactly matched to those used in this paper? If not, this should be clearly stated to ensure fair comparison.
>
> **A:** Yes, we used the official repos to conduct experiments. LESS, TSDS, and our method all select 10K samples and use these 10K samples to fine-tune the same model. The model is then evaluated on downstream tasks.
>
>
>
> > **Q4:** Have the authors analyzed the sensitivity of key parameters, such as the minimum score threshold or the number of clusters k? Without this, reproducibility and transferability could be limited.
>
> **A:** The minimum score threshold is analyzed in Table 3, which shows that when the minimum score is set to 7, aligning each label domain yields relatively stable performance. We have also conducted experiments on the number of clusters (k) in Figure 3 in our main paper, which demonstrates that performance is robust when k is in the range of 50 to 150.
>
> > **Q5:** The ablation only examines the combined effect of filtering and sampling. It would be helpful to further analyze how each stage contributes to different task types, such as reasoning, factual, or comprehension tasks.
>
> **A:** Thank you for your comments. Our ablation study in Table 5 is explicitly designed to analyze how each stage contributes to different task types, such as reasoning, factual, or comprehension tasks. We perform ablation studies across all datasets, rather than focusing on just one. Table 2 in our paper shows that each dataset corresponds to a different task type. The ablation studies Table 5 indicate that both filtering and sampling are crucial for efficient data selection.

---

### Official Review · Reviewer_RUUs · 2025-10-30

**Soundness:** 2
**Presentation:** 3
**Contribution:** 2
**Rating:** 4
**Confidence:** 4

**Summary:**

This paper proposes a proxy-label-based data selection method for instruction-tuning LLMs, aiming to select source data that best matches the target task by jointly considering instruction text and task-semantic proxy labels. The method targets limitations of prior work that only align input distributions without task semantics. Experiments across multiple benchmarks show improvements, though the gains vary by semantic field and threshold settings.

**Strengths:**

1. An interesting idea of jointly aligning instructional text and task-semantic proxy labels.
2. Comprehensive experimental coverage across multiple benchmarks and semantic dimensions, demonstrating systematic evaluation of the proposed approach.

**Weaknesses:**

1. The performance gains are inconsistent and not uniformly strong across benchmarks (Table 3). There is no single configuration that consistently outperforms others: for example, min-score ≥7 achieves two SOTA results, and min-score ≥6 also yields two SOTA results. Additionally, different semantic fields produce varying best configurations (e.g., TruthfulQA prefers “audience” under min-score ≥6 but “style” under ≥7), making it unclear how to select the semantic field and threshold in a principled manner. The authors also acknowledge inconsistent alignment effectiveness across fields (row 421), reinforcing this concern.
2. Important retrieval-style baselines such as representation-based RDS [1,2] and BM25 are missing, making it difficult to assess how much benefit comes from semantic distribution matching versus standard retrieval approaches.
3. The approach is modular and largely post-hoc rather than jointly optimized, which may limit conceptual novelty. The contribution appears to lie more in the combination of existing components.

[1] Zhang, R., Isola, P., Efros, A. A., Shechtman, E., and Wang, O. The unreasonable effectiveness of deep features as a perceptual metric. In *Proceedings of the IEEE conference on computer vision and pattern recognition*, pp. 586–595, 2018.

[2] Ivison, H., Zhang, M., Brahman, F., Koh, P. W., & Dasigi, P. (2025). *Large-Scale Data Selection for Instruction Tuning*. arXiv preprint arXiv:2503.01807.

**Questions:**

1. In Table 5, why does removing OOD filtering produce a very large drop on TruthfulQA.
2. Why is a separate OOD-filtering step required? Since Step 2 already computes similarity for anchor propagation, could OOD samples be filtered via a similarity threshold rather than a second LLM-based scoring step?
3. As a baseline or further exploration, what would happen if semantic-field information were integrated into existing data-selection approaches (e.g., adding semantic attributes to gradient-based LESS or representation-based RDS)? Would this mitigate the issue raised in rows 49-53 and unify the benefits without the need for proxy labeling and field-wise tuning?

---

> ### Author Response · Authors · 2025-11-25
> **Part-1 of Our Reply**
>
> Thank you for your valuable time and insightful comment. We have carefully read your comments and have responded and explained accordingly.
>
>
>
> > **W1:** The performance gains are inconsistent and not uniformly strong across benchmarks (Table 3). There is no single configuration that consistently outperforms others: for example, min-score ≥7 achieves two SOTA results, and min-score ≥6 also yields two SOTA results. Additionally, different semantic fields produce varying best configurations (e.g., TruthfulQA prefers “audience” under min-score ≥6 but “style” under ≥7), making it unclear how to select the semantic field and threshold in a principled manner. The authors also acknowledge inconsistent alignment effectiveness across fields (row 421), reinforcing this concern.
>
> **A:** Thank you for rising this concern. We acknowledge that the performance variations across benchmarks reflect a fundamental challenge in LLM data selection: A single method is hard to consistently dominate across all datasets. As shown in Table 3, even the baseline methods we compare against exhibit their own dataset-specific preferences. This pattern is also consistent with findings in task-unspecfic data selection, where methodological superiority tends to be context-dependent rather than universal.
>
> Our use of four label domains (task, topic, style, audience) primarily serves to highlight the flexibility of our framework. We apply these **general** label domains to different datasets. In practice, if we have little knowledge of the task, we can apply general label domains. However, if we have prior knowledge of the task, the choice of label domain can be guided by this knowledge. For example, in mathematical reasoning tasks, one could define the label domain as "mathematical problem-solving skills," as seen in [R1], which shows promise in improving exemplar quality using domain-specific labels. A more **explicit** domain definition can lead to more accurate data selection.
>
> As shown in Table 3 in our main paper, when applying **general** label domains ( task, topic, and style) to the five datasets, performance remains relatively stable when the minimum score exceeds 7 and our pipeline have strong performance on MMLU, TruthfulQA and GSM datasets compared to other task-specific data selection methods (LESS and TSDS).
>
> However, we acknowledge the importance of consistency in data selection. Our pipeline is an initial step in reformulating the joint alignment problem for task-specific data selection. We are committed to working on improving consistency in future iterations of our research.
>
> [R1] Metacognitive Capabilities of LLMs: An Exploration in Mathematical Problem Solving. NeurlPS 2024
>
>
>
> > **W2:** Important retrieval-style baselines such as representation-based RDS [1,2] and BM25 are missing, making it difficult to assess how much benefit comes from semantic distribution matching versus standard retrieval approaches.
>
> **A:** Thank you for this suggestion. We have added the requested baselines, and the results for BM25 and RDS+ are as follows:
>
> |                                          | MMLU | TruthfulQA | GSM  | BBH  | TyDiQA |
> | ---------------------------------------- | ---- | ---------- | ---- | ---- | ------ |
> | BM25                                     | 63.2 | 28.5       | 51.5 | 59.1 | 61.6   |
> | RDS+                                     | 63.6 | 3.5        | 54.0 | 59.1 | 60.2   |
> | Our pipeline (align topic; min score: 7) | 64.6 | 46.4       | 57.0 | 60.0 | 59.3   |
>
> The results indicate that our pipeline achieves superior performance across most tasks. We observe that while RDS+ performs well on MMLU, GSM, BBH, and TyDiQA, its performance on TruthfulQA is considerably lower (the original RDS+ paper does not report results on this dataset). These comparisons help illustrate the specific advantages of our approach relative to standard retrieval methods. We have included these results in Table 3 in our main paper.

---

> ### Author Response · Authors · 2025-11-25
> **Part-2 of Our Reply**
>
> > **W3:** The approach is modular and largely post-hoc rather than jointly optimized, which may limit conceptual novelty. The contribution appears to lie more in the combination of existing components.
>
> **A:** We appreciate your comment on the modular nature of our approach. Our primary contribution lies in the **reconceptualization of the task-specific data selection problem for instruction tuning (Section 3)**. Aligning $P(X, Y)$ between target and source without ground truth $Y$ presents a significant challenge, as it must address both label noise and domain (label) shifts, two distinct problems in machine learning. While it is possible to jointly optimize these issues, as in [R1], such an approach requires extensive model training, which is computationally expensive and reduces the efficiency of the selection process.
>
> Our current approach separates these two problems, avoiding the need for model fine-tuning, while still providing an efficient solution (Our running cost analyses is added to Appendix D.9). We agree that joint optimization could offer a more coherent method, and we view this as an exciting direction and will contine to pursue this avenue.
>
> [R1] Learning from Long-tailed Data with Noisy Label
>
>
>
> > **Q1:** In Table 5, why does removing OOD filtering produce a very large drop on TruthfulQA.
>
> **A:** The significant performance drop on TruthfulQA when OOD filtering is removed can be attributed to two main factors. First, as observed in other selection methods (e.g., the "Completion Length" method in Table 3), the Llama 3.1-8B model shows particular sensitivity to the data distribution for TruthfulQA. Second, without OOD filtering, the incremental sampling process struggles to reliably select data semantically close to the TruthfulQA due to substantial label noise in the source pool. Consequently, the model learns from noisier and less relevant data, leading to the significant performance drop.
>
>
> > **Q2:** Why is a separate OOD-filtering step required? Since Step 2 already computes similarity for anchor propagation, could OOD samples be filtered via a similarity threshold rather than a second LLM-based scoring step?
>
> **A:** Thank you for your insightful question. Yes, it is possible to filter OOD samples using a similarity threshold between the anchor and source samples, rather than using LLM-based scoring. However, the performance of this approach is less accurate compared to LLM-based judgment. For example, [R1] shows that LLM-based scoring provides more accurate and semantically meaningful results. We conducted additional experiments to replace our OOD filtering with a similarity threshold, while keeping all other modules the same:
>
> |                                             | MMLU | TruthfulQA | GSM  | BBH  | TyDiQA |
> | ------------------------------------------- | ---- | ---------- | ---- | ---- | ------ |
> | OOD filtering with similarity threshold     | 63.0 | 28.3       | 51.0 | 57.4 | 61.6   |
> | Our pipeline (OOD filtering with LLM-score) | 64.6 | 46.4       | 57.0 | 60.0 | 59.3   |
>
> As shown, LLM-based scoring outperforms the similarity threshold, particularly in terms of TruthfulQA performance. This experiment has been added to our main paper, Table 6.
>
> [R1] AlpaGasus: Training a Better Alpaca with Fewer Data. ICLR 2024
>
>
>
> > **Q3:** As a baseline or further exploration, what would happen if semantic-field information were integrated into existing data-selection approaches (e.g., adding semantic attributes to gradient-based LESS or representation-based RDS)? Would this mitigate the issue raised in rows 49-53 and unify the benefits without the need for proxy labeling and field-wise tuning?
>
>
>
> **A:** We thank the reviewer for this constructive question. To test whether integrating semantic-field information could mitigate the issue and unify the benefits, we conducted an experiment where we appended domain labels to the instructions and re-ran the RDS+ selection.
>
> |                                          | MMLU | TruthfulQA | GSM  | BBH  | TyDiQA |
> | ---------------------------------------- | ---- | ---------- | ---- | ---- | ------ |
> | RDS+                                     | 63.6 | 3.5        | 54.0 | 59.1 | 60.2   |
> | RDS+ (adding semantic-field information) | 63.7 | 5.4        | 53.5 | 60.7 | 59.7   |
> | Our full pipeline                        | 64.6 | 46.4       | 57.0 | 60.0 | 59.3   |
>
> The results demonstrate that this direct integration does not lead to a consistent or substantial improvement over the baseline RDS+. This supports our argument that a more structured strategy, such as explicit distribution alignment, is needed to effectively utilize semantic-field information for high-quality data selection.  We have put this experiment in Table 6 in our main paper.

---

### Official Review · Reviewer_MGtc · 2025-11-01

**Soundness:** 2
**Presentation:** 3
**Contribution:** 2
**Rating:** 4
**Confidence:** 2

**Summary:**

This paper reformulates task-specific data selection for LLM finetuning, arguing that prevailing methods, which only align the distribution of inputs X, are insufficient. The authors' central claim is that selection must instead align the joint distribution of inputs and labels $(X, Y)$ to capture true task relevance.

To achieve this, the paper proposes a novel four-stage pipeline that uses LLM-generated "proxy labels" since true labels are unavailable.

Experiments show that finetuning a LLaMA-3.1-8B model on a 10K subset selected with this method achieves performance competitive with or superior to state-of-the-art baselines and a model trained on the full 300K-sample pool.

**Strengths:**

1. This paper argues that task-specific data selection should not be based on aligning inputs ($X$) alone, which is the common practice, but on aligning the joint distribution of inputs and labels ($X, Y$). This is a more accurate and semantically meaningful way to define task relevance.
2. The paper introduces a novel four-stage pipeline that operationalizes its new formulation. Since target labels ($Y$) are typically unavailable, it uses an LLM to generate structured "proxy labels" (Task, Topic, Style, Audience). This provides a concrete and practical solution to the challenge of joint distribution matching.

**Weaknesses:**

1. The proposed 4-step pipeline is highly complex. It requires two distinct LLM-based steps (proxy-label generation and OOD scoring), an embedding model, k-means clustering, and an incremental sampling algorithm. This complexity introduces numerous hyperparameters that are not thoroughly justified, such as the number of clusters ($k=100$), the OOD score threshold, and the choice of which label field to align (Task, Topic, Style, or Audience), suggesting the method requires extensive, task-specific tuning to work well.
2. The ablation study in Table 5 does not adequately isolate the core contribution of the paper. The paper's main claim is that aligning the joint distribution $P(X, Y)$ is superior to aligning the marginal distribution $P(X)$. However, the ablation study only compares the full pipeline against removing its own components (OOD filtering or incremental sampling). A crucial missing baseline would be to apply the exact same clustering and incremental sampling algorithm (Steps 2 and 4) directly to the input embeddings ($X$) instead of the proxy-label embeddings ($Y$). Without this direct comparison, it is unclear if the performance gains come from the novel $P(X, Y)$ alignment or simply from the clustering/sampling algorithm itself.

**Questions:**

1. Your results in Table 3 demonstrate that the choice of which proxy label to align (e.g., "Align_task", "Align_topic", "Align_style") is a critical hyperparameter, as the best-performing field changes for each benchmark. For a practitioner applying your method to a new task, how would you recommend they determine the optimal field to align? Does this not require them to run multiple full finetuning experiments for each field, which would undermine the method's goal of data efficiency?
2. Your core claim is that aligning the joint distribution $P(X, Y)$ is superior to aligning the marginal input distribution $P(X)$. However, your ablation study in Table 5 only compares your full pipeline against versions with its own components (OOD filtering or incremental sampling) removed. To truly isolate the benefit of using proxy labels ($Y$), could you provide results for a baseline that applies your exact same pipeline (clustering, OOD filtering, and incremental sampling) but operates directly on the input instruction embeddings ($X$) instead of the proxy-label embeddings ($Y$)?

---

> ### Author Response · Authors · 2025-11-25
> **Part-1 of Our Reply**
>
> Thank you for your valuable time and insightful comment. We have carefully read your comments and have responded and explained accordingly.
>
>
>
> > **W1:** The proposed 4-step pipeline is highly complex. It requires two distinct LLM-based steps (proxy-label generation and OOD scoring), an embedding model, k-means clustering, and an incremental sampling algorithm. This complexity introduces numerous hyperparameters that are not thoroughly justified, such as the number of clusters (k), the OOD score threshold, and the choice of which label field to align (Task, Topic, Style, or Audience), suggesting the method requires extensive, task-specific tuning to work well.
>
> **A:** We explain our pipeline in terms of hyper-parameters robustness, choice of label domain, running time below:
>
> - **Hyperparameter Robustness:** We have conducted ablation studies on the number of clusters k, which are included in Figure 3 in the main paper. The results demonstrate that our method is robust to k values ranging from 50 to 150. Additionally, Appendix D.4 presents an analysis of our method's robustness to the quality of the proxy labels.
>
> - **Choice of Label Domain:** Our use of four label domains (task, topic, style, audience) primarily serves to highlight the flexibility of our framework. We apply these **general** label domains to different datasets. In practice, if we have little knowledge of the task, we can apply general label domain. However, if we have prior knowledge of the task, the choice of label domain can be guided by this knowledge. For example, in mathematical reasoning tasks, one could define the label domain as "mathematical problem-solving skills," as seen in [R1], which shows promise in improving exemplar quality using domain-specific labels. A more **explicit** domain definition can lead to more accurate data selection.
>
>   As shown in Table 3 in our main paper, when applying **general** label domains ( task, topic, and style) to the five datasets, performance remains relatively stable when the minimum score exceeds 7 and our pipeline have strong performance on MMLU, TruthfulQA and GSM datasets compared to other task-specific data selection methods (LESS and TSDS).
>
> - **Running Time and Complexity:** Although our pipeline involves multiple stages, it is computationally efficient compared to gradient-based baselines. We benchmark the running time on an A100 GPU  against two prominent methods: LESS and TSDS. Results are summarized below:
>
>
>
> |      | Stage-1                    | Stage-2                                   | Stage-3                     | Stage-4                     | Finetuning        | Total-time |
> | ---- | -------------------------- | ----------------------------------------- | --------------------------- | --------------------------- | ----------------- | ---------- |
> | LESS | Lora-training (6h)         | Gradient computation (51h)                | Data-selection (1 min)      |                             | Llama-3.1 8B (3h) | 60 h       |
> | TSDS | Lora-training (6h)         | Gradient computation (51h)                | KNN-KDE Data-selection (1h) |                             | Llama-3.1 8B (3h) | 61h        |
> | Ours | Target set annotation (1h) | Clustering and propagation to source (4h) | OOD filtering (11h)         | incremental-sampling (3min) | Llama-3.1 8B (3h) | 19h        |
>
>
>
> LESS is bottlenecked by gradient computation, as it requires model-specific gradients from the same architecture used for fine-tuning, and our reported runtime aligns with the original LESS paper. Similarly, TSDS inherits this substantial computational overhead by reusing the gradients computed by LESS, as acknowledged in Section 5.1 of the TSDS paper.
>
> In contrast, our method avoids expensive backward passes through large LLMs. The core of our efficiency comes from using the lightweight BGE-M3 model to extract embeddings for the entire source set. This step, which is essential for enabling the subsequent label propagation, takes only 4 hours. Target set annotation is fast due to its small scale. The overall pipeline, including clustering (4 minutes), propagation (4 hours), OOD filtering (11 hours), and incremental sampling (3 minutes), results in a total runtime of just 19 hours. Furthermore, by decoupling embedding extraction from the fine-tuning model, our approach remains model-agnostic, offering greater flexibility and scalability.
>
> This running-time analysis has been included in Appendix D.9 in our paper.
>
>
>
> [R1] Metacognitive Capabilities of LLMs: An Exploration in Mathematical Problem Solving. NeurlPS 2024

---

> ### Author Response · Authors · 2025-11-25
> **Part-2 of Our Reply**
>
> > **W2:** The ablation study in Table 5 does not adequately isolate the core contribution of the paper. The paper's main claim is that aligning the joint distribution $P(X,Y)$ is superior to aligning the marginal distribution $P(X)$ . However, the ablation study only compares the full pipeline against removing its own components (OOD filtering or incremental sampling). A crucial missing baseline would be to apply the exact same clustering and incremental sampling algorithm (Steps 2 and 4) directly to the input embeddings $X$ instead of the proxy-label embeddings $Y$ . Without this direct comparison, it is unclear if the performance gains come from the novel alignment $P(X,Y)$ or simply from the clustering/sampling algorithm itself.
>
> **A:** We thank the reviewer for this excellent suggestion. To directly isolate the benefit of aligning the joint distribution $P(X,Y)$ over the marginal distribution $P(X)$, we have conducted the proposed experiment. We applied our pipeline using only the input embeddings $X$ (i.e., without generating or using proxy labels $Y$). The results are shown below which demonstrate that aligning $P(X,Y)$ outperforms aligning only  $P(X)$, confirming that the performance gain is indeed attributable to the incorporation of proxy labels for joint distribution alignment, rather than just the clustering and sampling algorithm. We have added this experiment in the main paper, Table 6.
>
> |                               | MMLU | TruthfulQA | GSM  | BBH  | TyDiQA |
> | ----------------------------- | ---- | ---------- | ---- | ---- | ------ |
> | Aligning P(X) in our pipeline | 63.4 | 40.8       | 54.0 | 59.9 | 58.2   |
> | Aligning P(X,Y)               | 64.6 | 46.4       | 57.0 | 60.0 | 59.3   |
>
>
>
> > **Q1:** Your results in Table 3 demonstrate that the choice of which proxy label to align (e.g., "Align_task", "Align_topic", "Align_style") is a critical hyperparameter, as the best-performing field changes for each benchmark. For a practitioner applying your method to a new task, how would you recommend they determine the optimal field to align? Does this not require them to run multiple full finetuning experiments for each field, which would undermine the method's goal of data efficiency?
>
>
>
> **A:** Please refer to our response to **W1**.
>
>
>
> > **Q2:** Your core claim is that aligning the joint distribution $P(X, Y)$ is superior to aligning the marginal input distribution $P(X)$. However, your ablation study in Table 5 only compares your full pipeline against versions with its own components (OOD filtering or incremental sampling) removed. To truly isolate the benefit of using proxy labels ($Y$), could you provide results for a baseline that applies your exact same pipeline (clustering, OOD filtering, and incremental sampling) but operates directly on the input instruction embeddings ($X$) instead of the proxy-label embeddings ($Y$)?
>
> **A:** Plese Refer to our reply to  **W2**.

---

### Author Response · Authors · 2025-11-25
**Response to All Reviewers**

Dear AC and Reviewers,

We sincerely thank all reviewers for their thoughtful and constructive feedback. We are greatly encouraged that the reviewers found our core idea: framing task-specific data selection as a **joint alignment** problem to be novel, interesting, and a fresh perspective (MGtc, RUUs, dz9h, CwHM). We also appreciate the positive remarks on our comprehensive experiments (RUUs), coherent pipeline (dz9h), clear presentation (CwHM), and information-theoretic understanding (dz9h, MYpb).

We wish to emphasize that our primary contribution is the **reconceptualization of the task-specific data selection problem for instruction tuning (Section 3)**, rather than just the specific pipeline we present. Viewing it through the lens of joint distribution alignment unlocks a broader solution space, and our work serves as an initial instantiation of this perspective.

We have carefully considered all comments and revised the manuscript accordingly. Key updates, highlighted in blue in the manuscript, include:


- **Additional Baselines and Variants:** We incorporated new baseline comparisons (BM25, RDS+) and explored pipeline variants, such as using input-only embeddings and replacing our OOD filtering with cosine similarity ranking. We also evaluated the sensitivity of our method to the number of clusters k in K-means. These results are included in the main paper (Table 3, Table 6, and Figure 3).
- **Expanded Analysis:** In response to the reviewers’ valuable suggestions, we have added extensive analyses and experiments in Appendix D (Sections D.2 to D.8), covering LLM annotation quality, LLM score consistency, robustness to label noise, and more.


- **Efficiency Discussion:** We have added a detailed computational cost breakdown for each stage of our pipeline, including comparisons with input-only selection methods under the same hardware setup (Appendix D.9).

We have provided point-by-point responses to each reviewer’s comments below. We look forward to your reply and are happy to answer any further questions.

Thank you again for your valuable time and insights.

Best regards,

Authors of Submission 11624

---

### Meta-Review · Area_Chair_bg4d · 2026-01-07

**Summary:**

This paper addresses the problem of task-specific data selection for large language model (LLM) fine-tuning. The central claim is that prevailing approaches, which align only the input distribution, are insufficient, and that aligning the joint distribution of inputs and labels better captures task relevance. Since true labels are typically unavailable, the authors propose a four-stage pipeline that leverages LLM-generated proxy labels (Task, Topic, Style, Audience) to operationalize joint distribution matching. Experiments show that fine-tuning on subsets selected by this method can achieve performance competitive with or superior to state-of-the-art baselines and even models trained on full datasets.

**Reviewer Concerns:**

- The pipeline is complex, involving multiple LLM-based steps, clustering, and incremental sampling, with many hyperparameters that lack principled justification. This raises concerns about reproducibility and ease of adoption.
- The performance gains are inconsistent across benchmarks and depend heavily on choices of semantic field and threshold. There is no clear guidance for practitioners on how to select these parameters without extensive trial-and-error.
- The analysis of proxy label quality, consistency, and bias is limited. Without quantitative evaluation against human annotations, the reliability of proxy labels remains unclear.
- Efficiency and scalability are insufficiently discussed. The cost of generating proxy labels and running multi-stage selection is not broken down, leaving questions about real-world applicability.

**Reviewer Scores:**

Reviewers MYpb, MGtc, and RUUs have chances to raise their scores.

---

### Decision · Program_Chairs · 2026-01-26

Accept (Poster)